



# Power curve and wake analyses of the Vestas multi-rotor demonstrator

Maarten Paul van der Laan[1], Søren Juhl Andersen[2], Néstor Ramos García[2], Nikolas Angelou[1], Georg Raimund Pirrung[1], Søren Ott[1], Mikael Sjöholm[1], Kim Hylling Sørensen[3], Julio Xavier Vianna Neto[3], Mark Kelly[1], Torben Krogh Mikkelsen[1], and Gunner Christian Larsen[1]

[1]Technical University of Denmark, DTU Wind Energy, Risø Campus, Frederiksborgvej 399, 4000 Roskilde, Denmark
[2]Technical University of Denmark, DTU Wind Energy, Lyngby Campus, Anker Engelunds Vej 1, 2800 Lyngby, Denmark
[3]Vestas Wind System A/S, Hedeager 42, 8200 Aarhus, Denmark

**Correspondence:** Maarten Paul van der Laan (plaa@dtu.dk)

**Abstract.** Numerical simulations of the Vestas multi-rotor demonstrator (4R-V29) are compared with field measurements of power performance and remote sensing measurements of the wake deficit by a short-range WindScanner lidar system. The simulations predict a gain of 0-2% in power due to the rotor interaction, for wind speeds below rated. The power curve measurements also show that the rotor interaction increases the power performance below rated by $1.8\pm0.2\%$, which can result

in a $1.5\pm0.2\%$ increase in the annual energy production. The wake measurements and numerical simulations show four distinct wake deficits in the near wake, which merge into a single wake structure further downstream. Numerical simulations show that the wake recovery distance of a simplified 4R-V29 wind turbine is $1.03$-$1.44D_{\mathrm{eq}}$ shorter than for an equivalent single-rotor wind turbine with a rotor diameter $D_{\mathrm{eq}}$. In addition, the numerical simulations show that the added wake turbulence of the simplified 4R-V29 wind turbine is lower in the far wake compared to the equivalent single-rotor wind turbine. The faster wake

recovery and lower far-wake turbulence of such a multi-rotor wind turbine has the potential to reduce the wind turbine spacing within a wind farm while providing the same production.

## 1 Introduction

Over the last decades, the rated power of wind turbines has increased by upscaling the traditional concept of a horizontal axis wind turbine with a single three-bladed rotor. It is expected that this trend will continue for offshore wind turbines, although

the problems that arise from realizing large wind turbine blades (> 100 m) are not trivial to solve (Jensen et al., 2017). An alternative way to increase the power output of a wind turbine is the multi-rotor concept, where a single wind turbine is equipped with multiple rotors. From a cost point of view, it can be cheaper to produce a multi-megawatt wind turbine with several rotors consisting of relatively small blades that are already mass produced compared to a single-rotor wind turbine with newly designed large blades (Jamieson et al., 2014). In addition, small blades are easier to transport than large blades, which

makes a multi-rotor concept interesting for locations where the infrastructure is a limiting factor.

The multi-rotor wind turbine concept is an idea first introduced by Honnef (1932), as discussed by Hau (2013). In the late 20th century, the Dutch company Lagerwey Wind built and operated several multi-rotor wind turbine concepts based on two,



four and six two-bladed rotors (Jamieson, 2011). In April 2016, Vestas Wind Energy Systems A/S built a multi-rotor wind turbine demonstrator at the Risø Campus of the Technical University of Denmark. This multi-rotor wind turbine, hereafter abbreviated as the 4R-V29 wind turbine, consists of four V29-225 kW rotors, laid out in a bottom and top pair. The 4R-V29 wind turbine has been operated for almost three years and was decommissioned in December 2018. In the present article, we

investigate the power performance and wake interaction of the 4R-V29 wind turbine using field measurements and numerical simulations.

The tip clearances between the rotors in multi-rotor wind turbines are typically much smaller than a single rotor diameter, and several authors have shown that the operating rotors strongly interact with each other. Chasapogiannis et al. (2014) and Jamieson et al. (2014) have performed numerical simulations of closely spaced rotors positioned in a honeycomb layout

with a tip clearance of 5% of the (single) rotor diameter. Chasapogiannis et al. (2014) simulated seven 2 MW rotors using computational fluid dynamics and vortex models, and they calculated an increase in power and thrust of about 3% and 1.5%, respectively, compared to seven non-interacting single rotors. In addition, Chasapogiannis et al. (2014) found that the seven individual single wakes merge into a single wake structure after a downstream distance of two rotor diameters. Jamieson et al. (2014) have simulated a 20 MW multi-rotor wind turbine consisting of 45 444 kW rotors, and argued that wind turbine loads

are reduced and the power performance is increased compared to a single 20 MW rotor. It is not clear how these results are calculated because their simulations are based on a blade element moment model, which cannot capture the aerodynamic interaction of the closely spaced rotors without additional modeling.

Ghaisas et al. (2018) have employed large-eddy simulations (LES) and two engineering wake models to show that a general multi-rotor wind turbine consisting of four rotors has a faster wake recovery and lower turbulent kinetic energy in the wake

compared to a single rotor with an equivalent rotor area. They argued that the faster wake recovery is a result of a larger entrainment because the ratio of the rotor perimeter and the rotor swept area is twice as high for their multi-rotor turbine compared to a single-rotor turbine. In the same work, it was also shown that different tip clearances in the range of zero to two rotor diameters hardly effect the wake recovery of the multi-rotor wind turbine, while the turbulent kinetic energy in the far wake varies, although its always less than the turbulent kinetic energy in the far wake of a single rotor. Finally, it was shown that

the power deficit and the added turbulent kinetic energy in the wake of a row of five multi-rotor wind turbines is less compared to a row of five single-rotor wind turbines. These results suggest that a wind farm of multi-rotors has less power losses and fatigue loads due to wakes compared to a wind farm of single-rotor wind turbines. In present article, we would like to confirm the results of Ghaisas et al. (2018) for the 4R-V29 wind turbine using different models and levels of ambient turbulence.

## 2   Field measurements

### 30   2.1   Definition of 4R-V29 wind turbine

Fig. 1 depicts the 4R-V29 wind turbine located at the Risø Campus of the Technical University of Denmark and a corresponding sketch including dimensions and rotor definitions. The hub height of the bottom rotor pair ($R_1$ and $R_2$) and the top rotor pair ($R_3$ and $R_4$) are 29.04 and 59.50 m, respectively, which gives an average hub height of 44.27 m. The horizontal distance



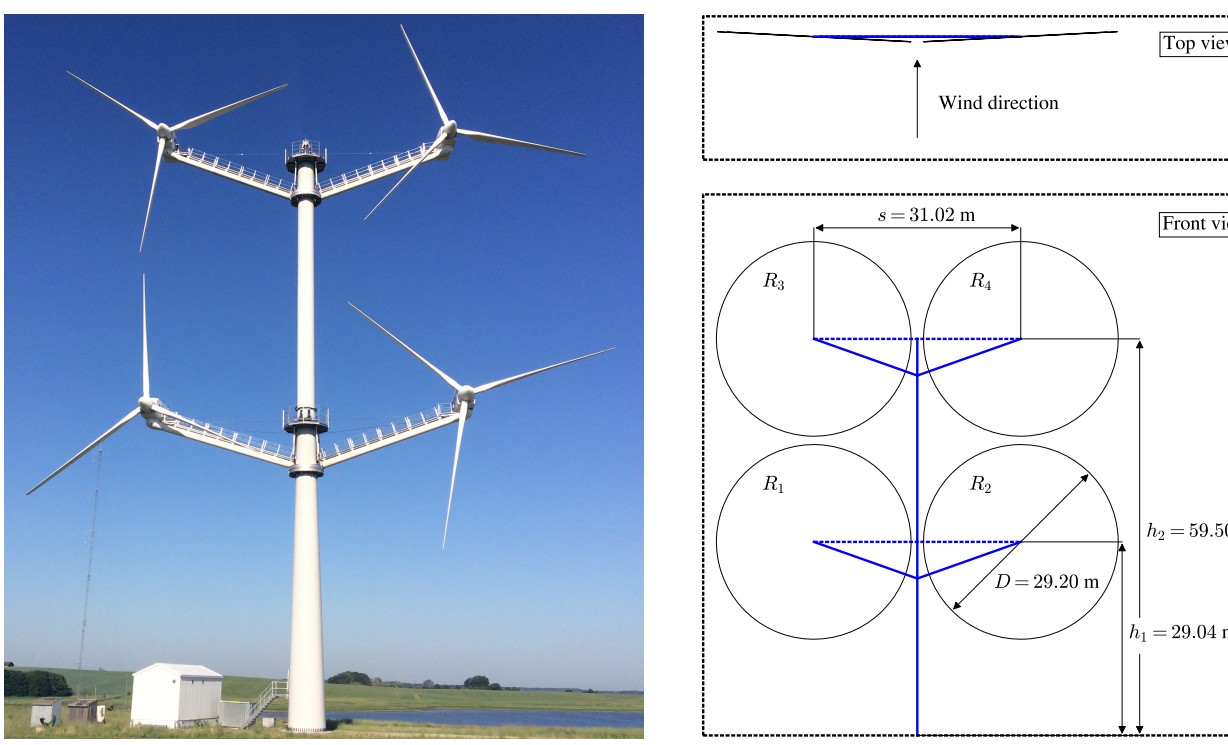

**Figure 1.** Left: 4R-V29 wind turbine located at the Risø Campus. Right: sketch of 4R-V29 wind turbine including dimensions and rotor definitions shown in a top and front view.

between the nacelles for both pairs is 31.02 m. The rotors are equipped with 13 m (V27) blades, where the blade root is extended by 1.6 m, resulting in a rotor diameter of 29.2 m. The rotor tilt angles and the cone angles (the angle between the individual rotor plane and its blade axis) are all zero. To increase the horizontal distance between the rotors (tip clearance), the 4R-V29 has a toe-out angle of 3°, as depicted in the top view sketch of Fig. 1. This means that the left rotors ($R_1$ and $R_3$) and right

5 rotors ($R_2$ and $R_4$) are yawed by +3° and -3°, respectively. (A positive yaw angle is a clockwise rotation as seen from above.) The horizontal and vertical tip clearances are 1.86 and 1.26 m, or 6.4% and 4.3% of the single rotor diameter, respectively, which is close to the 5% that has been used in simulations performed by Chasapogiannis et al. (2014) and Jamieson et al. (2014). It is possible to yaw the bottom and top pairs independently from each other, which could be beneficial in atmospheric conditions where a strong wind veer is present (i.e. a stable atmospheric boundary layer).

10 **2.2 Power curve measurements**

Power curve measurements of the 4R-V29 wind turbine were carried out to quantify the effect of the rotor interaction on the power performance. For this purpose, a test cycle of three stages has been run repetitively, as illustrated in Fig. 2. During Stages





and 3, only rotors $R_1$ and $R_3$ are in operation, respectively, while all other rotors are in idle. During Stage 2, all rotors are in operation. We use two single-rotor operation stages to account for the effect of the shear. Each stage is run for 15 min and are post processed to 10 min data samples by the removal of start up and shutdown periods between the stages. By toggling the stages at every 15 min, we minimize differences in environmental conditions between the three data sets (one data set per

stage).

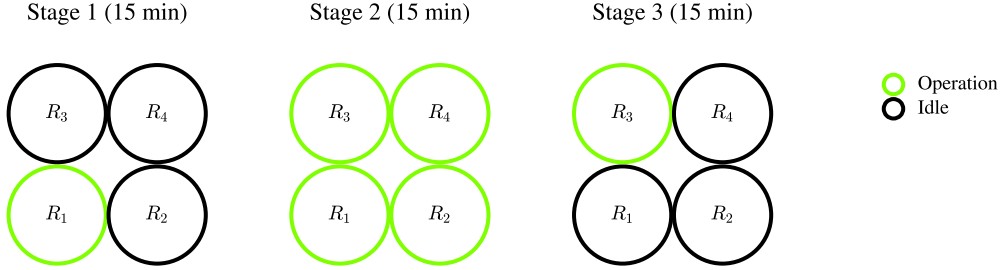

**Figure 2.** Test cycle of power curve measurements of 4R-V29 wind turbine.

The reference wind speed is measured by a commercial dual mode continuous wave lidar, ZephIR 300, manufactured by ZephIR (UK) (Medley et al., 2014). The lidar is mounted on the top platform of the 4R-V29 wind turbine at height of 60 m, as depicted in the left image of Fig. 3. It measures the upstream wind speed at 146 m ($5D$) and 300 m ($\approx 10.3D$), at a height of 44.3 m, as shown in the right plot of Fig. 3. We choose to use the lidar measurements of the reference wind speed at 146

m because the lidar measurements at 300 m have less data availability and a higher volume averaging. In order to capture the wind speed at a hub height of 44.3 m, it is configured with a tilt angle of -7 deg, such that the center of the scan is directed towards the desired measurement height, as illustrated in Fig. 3. A horizontal pair of measurements at this height are used to determine the wind speed and yaw misalignment, using a pair-derived algorithm. The lidar measurements are corrected in real time for tilt variations due to the tower deflection. A sample, measured every 0.05 s (20 Hz), is corrected for the difference in

the induction zone for when only one or all four rotors are in operation, as discussed in Sect. A. The corrected data samples are averaged over 10 minutes and then binned in wind speed intervals of 0.5 m/s.

The total available measurement cycles is depicted in the right plot of Fig. 4 and corresponds to 549 10 min data samples or approximately 91.5 h for each stage between wind speeds of 4 and 14 m/s. The total amount of data per stage is about half of the minimal requirement as defined in the international standard (IEC), where a power curve database should include at least 180

h of data and a minimal of 30 min per binned wind speed. In addition, there is not much data available above rated. As a result, the standard error of the mean power in a bin is high, as shown by the error bars in the power curves of Fig. 4 (left plot). These two power curves represent the sum of power from rotors $R_1$ and $R_3$ of Stages 1 and 3, and the sum of power from rotors $R_1$ and $R_3$ of Stage 2, both multiplied by a factor two. The relative difference between the power curves is discussed in Sect. 4.2. The power curve measurements are filtered for events, where a rotor (that is planned to operate) is not in full operation. During

the power curve measurements, the neighbouring V27 and Nordtank (NTK) wind turbines were not in operation (see Fig. 5). To avoid the influence of the other neighboring wind turbines and flow disturbance of a motorway, the power measurements




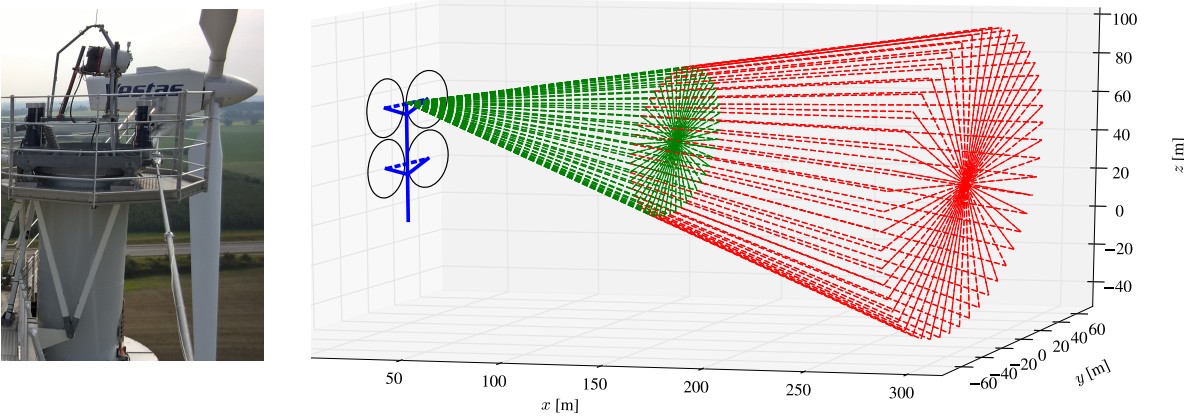

**Figure 3.** Left: ZephIR Z300 placement on the 4R-V29 wind turbine. Right: Scanning cone for $x = 146$ m (green) and $x = 300$ m (red).

are filtered for a wind direction sector 180-330°, which represents an inflow from the fjord (see Fig. 5). It should be mentioned that the wind turbine test site of the Risø Campus is not flat. The influence of this is minimized by adjusting the lidar configured height to match the height difference upstream, though this could give a slight influence on the power curve measurements. In addition, the power curve measurements are not filtered for turbulence intensity and atmospheric stability. However, the measurements are filtered for normalized mean fit residual below 4%, which removes data samples with high complexity of incoming flow.

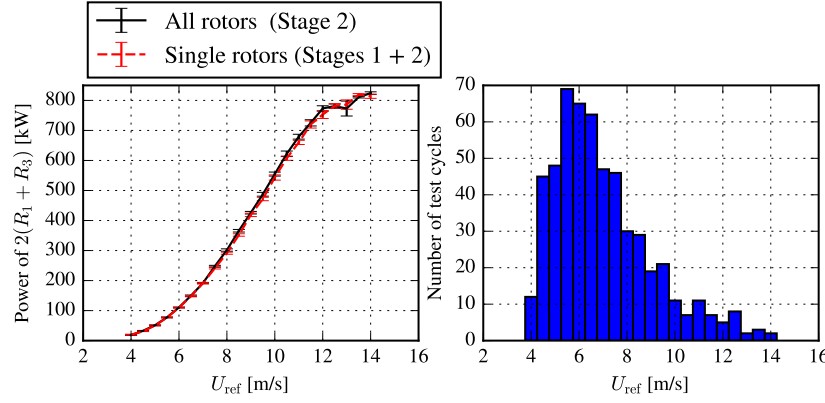

**Figure 4.** Left: Measured electric power of $2(R_1 + R_3)$ from the 4R-V29 wind turbine as function of the freestream velocity at a height of 44.27 m for all rotors and single rotors in operation. Error bars represents the standard error of the mean power. Right: Number of test cycles.



## 2.3 Wake measurements

The wake of the 4R-V29 turbine has been measured by three ground based short-range WindScanners (Mikkelsen et al., 2017; Yazicioglu et al., 2016) in two separate measurement campaigns, referred as the near wake and the far wake. The measurement setup is shown in Fig. 5. The three WindScanners measure the wake deficit by synchronously altering the line-of-sight azimuth

and elevation of each individual unit. In the near wake campaign, the WindScanners are scanning three cross planes located at $0.5D$, $1D$ and $2D$ downstream. In addition a horizontal line at the lower hub height $1D$ downstream was rapidly scanned at about 1 Hz. Each cross plane/line is scanned for 10 min, before moving on to the next, which means that every 40 min a complete set of three cross planes is available. The data is stored in one-minute files and the 10 min scans were post processed for minutes without scan plane transitions, rendering 8 min means. The far wake campaign consists of only one cross plane

scanned at $5.5D$ downstream. It is not possible to scan further downstream because of a highway and surrounding trees located at 170-200 m downstream of the 4R-V29 wind turbine for a wind direction of 280°. The WindScanners are positioned in between the near and far wake scanning distances. The selected WindScanner positions allow the monitoring of the near and far wake measurements by turning the pointing direction to and away from the 4R-V29 wind turbine, respectively. Such a configuration allows the estimation of the two components of the horizontal wind vector by assuming that the vertical

component was equal to zero. During the wake measurements, the neighboring Nordtank (NTK) and V27 wind turbines were not in operation.

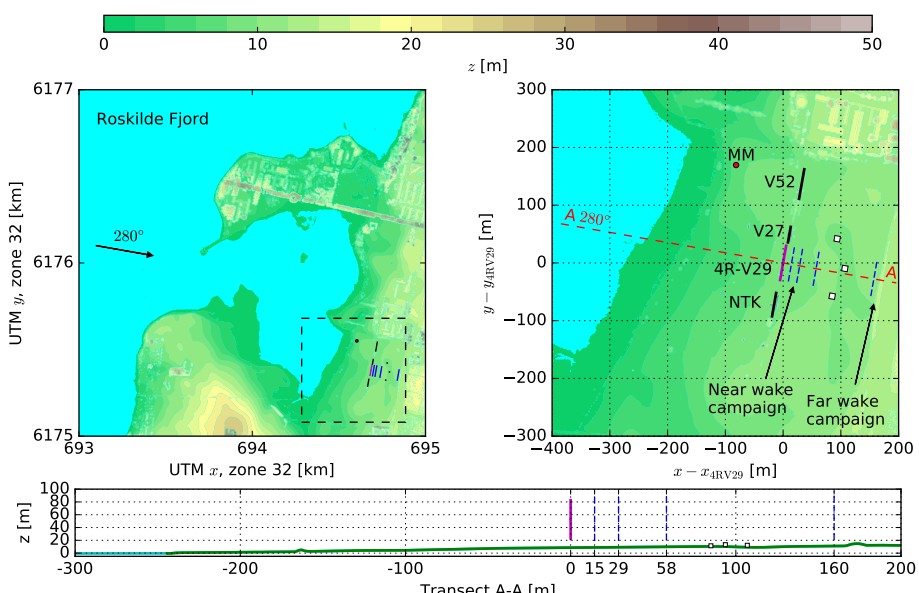

**Figure 5.** Topography around the 4R-V29 wind turbine and overview of wake measurements. Right top plot is a zoom of the left top plot. Bottom plot is the terrain height along transect A-A. Location of three WindScanners are shown as white squares, scanning locations are shown as blue dashed lines, MM is the met mast, and V52, V29 and NTK are neighboring wind turbines.




Fig. 6 summarizes the atmospheric conditions during the near and far wake measurements, as measured at the met mast depicted in Fig. 5. The met mast is equipped with pairs of cups and sonic anemometers located at five heights: 18, 31, 44, 57 and 70 m. The wind speed and wind direction are taken from a cup and a sonic, respectively, both located at a height of 44 m, which is close to the average hub height of the 4R-V29 wind turbine. The turbulence intensity and the atmospheric stability in

the terms of a Monin Obukhov length $L_{\mathrm{MO}}$ are measured by sonics located at a height of 44 and 18 m, respectively.

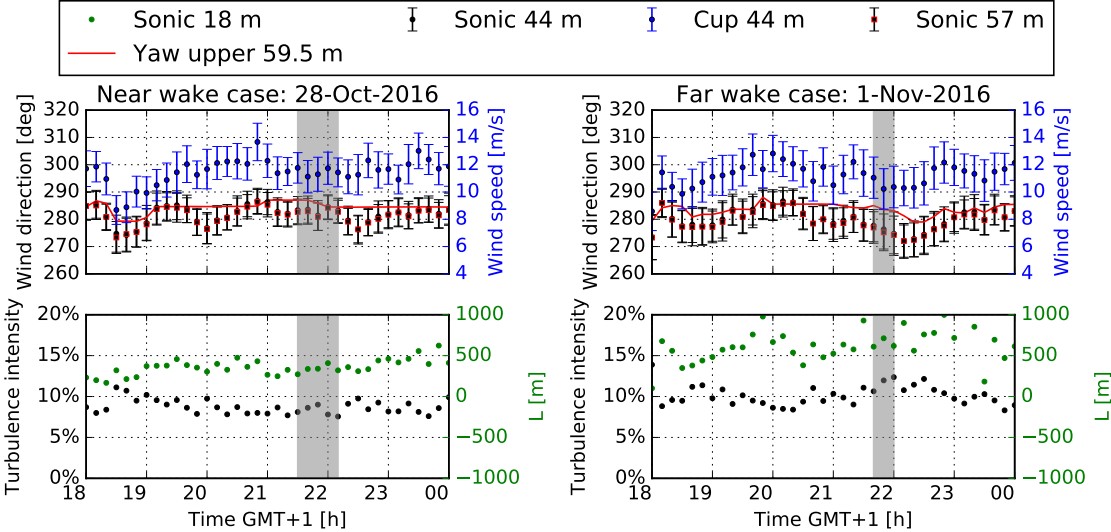

**Figure 6.** Atmospheric conditions during the near and far wake measurements as measured at V52 met mast. Top plots: Wind direction and wind speed and yaw angle of the upper platform. Bottom plots: Total turbulence intensity and Monin Obukhov length. Error bars represent the standard error of the mean. Gray area depict the time of the chosen measurements.

A near wake case is selected from three consecutive post processed scans measured between 21:36 and 22:03 GMT+1 on 28-Oct-2016. A far wake case is taken from one post processed scan measured between 21:45 and 21:53 GMT+1 on 1-Nov-2016. During these periods, the atmospheric stability is near neutral ($L_{\mathrm{MO}} = 340$ m) and neutral ($L_{\mathrm{MO}} = 661$ m). The wind direction in both cases is close to $280°$, and the yaw offset with respect to the upper platform is $3.4°$and $8.2°$for the near and

far wake cases, respectively. The atmospheric conditions of the two cases are listed in Tab. 1, which are used as input for the numerical simulations.

The wind speed and total turbulence intensity profiles measured at the met mast during the near and far wake case recordings are depicted in Fig. 7. The wind speed and turbulence intensity at 44 m ($U_{\mathrm{ref}}$ and $I_{\mathrm{ref}}$) are used to determine neutral logarithmic inflow profiles defined by $z_0$ and $u_*$ following Eq. (2). The results are listed in Tab. 1. The far wake profile deviates from a

logarithmic profile at a height of 18 m, which could be related to the upstream fjord-land roughness changes, as shown in Fig. 5, although this deviation is not observed in the near wake case inflow profile.

Spectra of 35 Hz wind velocity data measured by the sonic anemometer at 44 m are used to fit Mann turbulence spectra (Mann, 1994) using three parameters: $\alpha\epsilon^{\frac{2}{3}}$, $L$ and $\Gamma$. When these parameters are used to generate a Mann turbulence box that




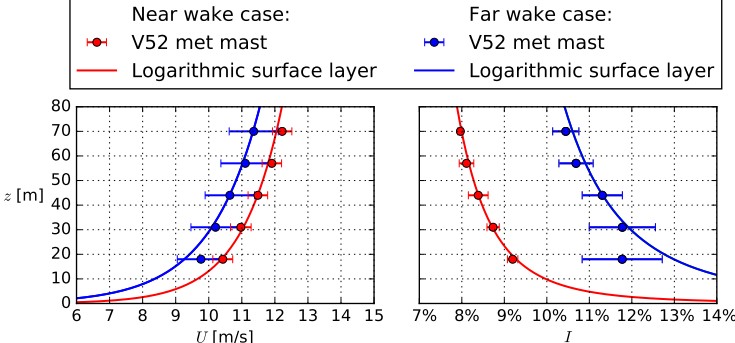

**Figure 7.** Profiles of wind speed and turbulence intensity measured at the met mast and corresponding logarithmic surface layer using $U_{\mathrm{ref}}$ and $I_{\mathrm{ref}}$ from Tab. 1 for the near and far wake measurement cases. Error bars represent the standard error of the mean.

is employed as inflow turbulence for the MIRAS-FLEX5 and EllipSys3D LES-AL-FLEX5 simulations (Sect. 3), the resulting turbulence intensity in the Mann turbulence box is lower than the measured value at the sonic anemometer, which is not fully understood. The problem is circumvented by using an $\alpha\epsilon^{\frac{2}{3}}$ that is about twice as large as original fitted. The final values of $\alpha\epsilon^{\frac{2}{3}}$, $L$ and $\Gamma$ are listed in Table 1. Note that the stream-wise dimension of the Mann turbulence box is chosen to fit an entire

5     measurement case (40 min) using $2^{14}\times2^{7}\times2^{7}$ points in the stream-wise and cross direction, respectively, with a spacing of 2 m in all directions.

**Table 1.** Summary of test cases based on wake measurements and corresponding input parameters for numerical computations.

| Case | Wake measurements | Directly measured | | | Logarithmic profile Eq. (2) | | Mann model fit | | |
|---|---|---|---|---|---|---|---|---|---|
| | | $U_{\mathrm{ref}}$ | $I_{\mathrm{ref}}$ | $L_{\mathrm{MO}}$ | $u_*$ | $z_0$ | $\alpha\epsilon^{\frac{2}{3}}$ | $L$ | $\Gamma$ |
| | | [m/s] | [%] | [m] | [m/s] | [m] | [$m^{4/3}s^{-2}$] | [m] | [-] |
| Near wake | $0.5D, 1D, 2D$ | 11.5 | 8.4 | 340 | 0.492 | $3.88\times10^{-3}$ | 0.086 | 41.8 | 3.34 |
| Far wake | $5.5D$ | 10.6 | 11.3 | 661 | 0.611 | $4.27\times10^{-2}$ | 0.1 | 47.7 | 4.37 |

## 3    Simulation methodology

Four different simulations tools are employed to model the 4R-V29 wind turbine: Fuga, EllipSys3D RANS-AD, MIRAS-FLEX5 and EllipSys3D LES-AL-FLEX5. The simulation methodology for each model, ranking from the lowest to highest

10     model fidelity, is described in the proceeding sections. Note that a high model fidelity corresponds to an intended high accuracy for the price of a high computational cost, although a good model performance is not guaranteed. All simulations that are used to model the 4R-V29 wind turbine only assume a neutral atmospheric surface layer inflow. In addition, only flat terrain with





a homogeneous roughness length is modeled, hence the effects of the fjord-land roughness change and sloping terrain are neglected.

## 3.1 Fuga

Fuga is fast linearized RANS model developed by Ott et al. (2011). Fuga models a single wind turbine wake as a linear
perturbation of an atmospheric surface layer. In the present setup, a thrust force is modeled that is distributed uniformly over the rotor swept area. The forces are smeared out using a two dimensional Gaussian filter with standard deviations of $D/4$ and $D/16$ in the stream-wise and cross directions. The turbulence is closed by taking the eddy viscosity of an atmospheric surface layer, which means that a wind turbine wake does not affected the turbulent mixing. The resulting equations are transformed to wave number space in the horizontal directions to obtain a set of mixed spectral ordinary differential equations. Since these
equations are very stiff, a novel numerical solving method is developed by Ott et al. (2011). The linearity of the model allows the superposition of single wakes, also in multi-rotor configurations.

## 3.2 EllipSys3D RANS-AD

EllipSys3D is an incompressible finite volume flow solver initially developed by Sørensen (1994) and Michelsen (1992), and has both RANS and LES models, and different methods to represent a wind turbine. In this section, the RANS-AD method is
discussed. The Navier-Stokes equations are solved with a SIMPLE algorithm (Patankar and Spalding, 1972), and the convective terms are discretized with a QUICK scheme (Leonard, 1979). The wind turbine rotors are represented by actuator disks (AD) based on airfoil data as presented in Réthoré et al. (2014). The RANS-AD model can only model stiff blades. The tip correction of Pirrung and van der Laan (2018) is applied (with a constant of $c_2 = 29$), which is an improved tip correction of Shen et al. (2005). This modified tip correction models the induced drag due to the tip vortex, which leads to a stronger tip loss effect
on the in-plane than on the out-of-plane forces. The RANS-AD model can be employed to model two different flow cases, a uniform inflow and a neutral atmospheric surface layer, which are described in the following Sects. 3.2.1 and 3.2.2. The uniform inflow case is used to validate the AD model of a single V29 rotor with the results of two blade element moment codes. The neutral atmospheric surface layer flow case is used to simulate the 4R-V29 wind turbine.

### 3.2.1 Uniform inflow case

For the uniform inflow case, the numerical setup is fully described in Pirrung and van der Laan (2018). The uniform grid spacing around the AD is set to $D/20$, which is fine enough to estimate $C_T$ and $C_P$ within a discretization error of 0.3% following a previously performed grid refinement (Pirrung and van der Laan, 2018).

### 3.2.2 Atmospheric surface layer flow case

For the atmospheric surface layer flow case, the $k$-$\varepsilon$-$f_P$ model from van der Laan et al. (2015) is employed, which is a modified
$k$-$\varepsilon$ model developed to simulate wind turbine wakes in atmospheric turbulence. A typical numerical domain for ADs in flat





terrain and corresponding boundary conditions are employed as described in van der Laan et al. (2015). In the present work, a finer spacing of $D/20$ is applied (in previous work of van der Laan et al. (2015) a spacing of $D/10$ has been used), and a larger uniformly spaced wake domain is used: $15D \times 5D \times 4D$ (stream-wise, lateral and vertical directions), where $D$ is a single rotor diameter, and the 4R-V29 wind turbine is placed at $3D$ downstream from the start of the wake domain. In addition, a larger outer domain is used; $116D \times 105D \times 50D$, such that the blockage effects are negligible (blockage ratio: $\pi/(105 \times 50) = 0.06\%$). In the RANS simulations, we have observed that a blockage ratio of 1% for the 4R-V29 wind turbine is not enough when comparing the simulated power of the 4R-V29 wind turbine with a single V29 rotor using the same domain. This is because the blockage ratio of the single V29 rotor simulation is four times lower than the 4R-V29 wind turbine simulation, and one would include a false gain in power for the 4R-V29 wind turbine that is caused by the difference in blockage ratio between the V29 and 4R-V29 wind turbine RANS simulations.

The inflow conditions represent a neutral atmospheric surface layer that is in balance with the domain (without the ADs):

$$U = \frac{u_*}{\kappa} \ln\left(\frac{z + z_0}{z_0}\right), \qquad k = \frac{u_*^2}{\sqrt{C_\mu}}, \qquad \varepsilon = \frac{u_*^3}{\kappa(z + z_0)} \tag{1}$$

where $U$ is the stream-wise velocity, $u_*$ is the friction velocity, $\kappa = 0.4$ is the Von Kármán constant, $z$ is the height, $z_0$ is the roughness length, $k$ is the turbulent kinetic energy, $C_\mu = 0.03$ the eddy viscosity coefficient, and $\varepsilon$ is the turbulent dissipation. The friction velocity and the roughness height can be set using a reference velocity $U_{\text{ref}}$ and a reference (total) turbulence intensity $I_{\text{ref}} = \sqrt{\frac{2}{3}k}/U_{\text{ref}}$, for a reference height $z_{\text{ref}}$:

$$u_* = U_{\text{ref}} I_{\text{ref}} \frac{C_\mu^{1/4}}{\sqrt{2/3}}, \qquad z_0 = \frac{z_{\text{ref}}}{\exp\left(\frac{\kappa\sqrt{2/3}}{I_{\text{ref}} C_\mu^{1/4}}\right) - 1} \tag{2}$$

The shear exponent from the power law ($U = U_{\text{ref}}(z/z_{\text{ref}})^\alpha$) can be expressed by setting the shear at the reference height ($\partial U/\partial z|_{z_{\text{ref}}}$) from the power law equal to that from the logarithmic profile and substituting Eq. (2):

$$\alpha = \frac{u_*}{\kappa U_{\text{ref}}} \frac{z_{\text{ref}}}{(z_{\text{ref}} + z_0)} = I_{\text{ref}} \frac{C_\mu^{1/4}}{\kappa\sqrt{\frac{2}{3}}} \left[1 - \exp\left(-\frac{\kappa\sqrt{\frac{2}{3}}}{I_{\text{ref}} C_\mu^{1/4}}\right)\right] = 1.274 I_{\text{ref}} + \mathcal{O}\left(I_{\text{ref}}^2\right) \tag{3}$$

Note that the power law is not used in the simulations; however, the relation in Eq. 3 is employed to discuss the simulations in Sect. 4.1.

### 3.3 MIRAS-FLEX5

The in-house solver MIRAS, Method for Interactive Rotor Aerodynamic Simulations, is a multi-fidelity computational vortex model for predicting the aerodynamic behavior of wind turbines and the corresponding wakes. It has been developed at the Technical University of Denmark during the last decade, and it is extensively validated for small to large size wind turbine rotors by Ramos-García et al. (2014a, b, 2017). The turbine aeroelastic behavior is modeled by using the MIRAS-FLEX5 aeroelastic coupling developed by Sessarego et al. (2017). FLEX5 is an aero-elastic tool developed by Øye (1996), which gives loads and deflections.



In the present study, a lifting line technique is employed as the blade aerodynamic model. The blade bound circulation is modeled by a vortex line, located at the blade quarter-chord and sub-divided in vortex segments. The vorticity is released into the flow by a row of vortex filaments following the chord direction (shed vorticity, which accounts for the released vorticity due to the time variation of the bound vortex) and a row of filaments perpendicular to the chord direction (trailing vorticity, which accounts for the vorticity released due to circulation gradients along the span-wise direction of the blade).

A hybrid vortex method is used for the wake modelling, where the near wake is modelled with vortex filaments and further downstream the filaments circulation is transformed into a vorticity distribution on a uniform Cartesian auxiliary mesh, where the interaction is efficiently calculated by an FFT-based method developed by Hejlesen (2016). Effects of domain blockage are removed by solving the Poisson equation using a regularized Green's function solution with free-space boundary conditions in all directions except the ground, which is modeled using a slip wall. In order to avoid the periodicity of the Green's function convolution, the free-space boundary conditions are practically obtained by zero-padding the domain, as introduced by Hockney and Eastwood (1988). The ground condition is modelled by solving an extended problem, accounting for the vorticity field mirrored about the ground plane.

The prescribed velocity-vorticity boundary layer model (P2VBL) presented in Ramos-García et al. (2018) is employed to model the wind shear. This model corrects the unphysical upward deflection of the wake observed in simpler prescribed velocity shear approaches.

The Mann model (Mann, 1998) is used to generate a synthetic turbulent velocity field on a uniform mesh, commonly known as a turbulence box. The velocity field is transformed into a vortex-particle cloud, which is gradually released into the computational domain at a plane $2D$ upstream the wind turbine. All components of Mann model velocity fluctuations are scaled by a factor 1.2 in order to reproduce the measured turbulence intensity at hub height (as listed in Tab. 1). It is not fully understood why this scaling factor is necessary in order to reproduce the original inflow turbulence intensity, and it should be investigated further in future work.

The used mesh has an extent of $L_x \times L_y \times L_z = 17.1D \times 6.2D \times 6.2D$, where $L_x$, $L_y$ and $L_z$ is the stream-wise, lateral and vertical domain length, respectively. A constant spacing of $0.7\,\mathrm{m}$, approximately 20 cells per blade, is used in all three directions, resulting in a mesh with $714 \times 258 \times 258$ cells. This sums up to a total of about 48 million cells with a number of vortex particles in the same order. Due to aeroelastic constraints the time step is fixed to 0.01s. A total number of 70000 time steps have been simulated for all cases. The analysis performed in the following sections use the data recorded for the last 60000 time steps. The turbulent box used in all computations is much larger than the actual simulated domain, $1122D \times 9D \times 9D$, in order to include large structures in the simulation. Moreover, the discretization of the box is coarser, with a constant spacing of $2\,\mathrm{m}$, which is around 3 times larger than the computational cells. In this way the smaller turbulent structures are generated by the solver.





### 3.4 EllipSys3D LES-AL-FLEX5

The structure of EllipSys3D is similar to that described in Sect. 3.2. For the LES cases the convective terms are discretized through a combination of the third order QUICK scheme and fourth order central difference scheme in order to suppress unphysical numerical wiggles and diffusion. The pressure correction equation is solved using the PISO algorithm.

LES applies a spatial filter on the Navier-Stokes equations, which yields in a filtered velocity field. The large scales are solved directly by the Navier-Stokes equations, while scales smaller than the filter scale is modeled using a sub-grid scale (SGS) model, which provides the turbulence closure. The SGS model is a mixed scale model based on an eddy-viscosity approach as described by Ta Phuoc et al. (1994).

  The turbines are modeled using the actuator line(AL) technique as described by Sørensen and Shen (2002), which applies
body forces along rotating lines within the numerical domain of the flow solver, here EllipSys3D. The body forces are computed using FLEX5. The actuator lines are therefore controlled directly by FLEX5, which means that the actuator lines are not only rotating, but also deflecting within the flow. Additional details of the aero-elastic coupling can be found in Sørensen et al. (2015). The aero-elastic coupling also provides a turbine controller, which is made up of a variable speed P-controller for below rated winds and a PI-pitch angle controller for wind speeds above rated, see Larsen and Hanson (2007) or Hansen et al.
(2005) for details on turbine controllers.

  The atmospheric boundary layer is modeled by applying body forces throughout the domain, see Mikkelsen et al. (2007). Applying body forces makes it possible to impose any vertical velocity profile, which is beneficial when aiming to model specific measurement, e.g. Hasager et al. (2017).

  Turbulence has also been introduced $2D$ upstream the turbines using body forces, see e.g. Gilling et al. (2009), where the
imposed turbulence is identical to the turbulence generated using the Mann model as described in Sect. 3.3. All components of Mann model velocity fluctuations are scaled by a factor 1.2 in order to reproduce the measured turbulence intensity at the wind turbine position, at hub height (as listed in Tab. 1).

  The computational mesh is $L_x \times L_y \times L_z = 17.5D \times 7D \times 20D$ in the stream-wise, lateral and vertical directions, respectively. This yields a blockage ratio of $2\%$, which is less that than $3\%$ as recommended by Baetke et al. Baetke et al. (1990). The mesh
is equidistant in the streamwise direction and in a region containing the turbine and wake of $2 - 6D$ in the lateral and from the ground up to $4D$ in the vertical, which is then stretched towards the sides. This corresponds to each turbine blade being resolved by 36 cells in order to resolve the tip vortices, Troldborg (2008), and the mesh contains a total of 131 million cells. Inlet and outlet boundary conditions have been applied in the streamwise direction, and cyclic boundary conditions on the lateral. The top boundary is modeled as a symmetry condition and the ground with a no-slip condition.

The simulations have been run with time steps of 0.0063s and 0.0069s for the near and far wake case, respectively.

  The presented statistics are based on 10 minutes, which have been sampled after the initial transients have propagated through the domain, similar to the results using MIRAS-FLEX5.





## 4   Results and Discussion

### 4.1   Comparison of V29 rotor models

A comparison of the V29 rotor models from EllipSys3D RANS-AD and FLEX5 (used by EllipSys3D LES-AL-FLEX5 and MIRAS-FLEX5) is made with a HAWC2 model of the V29 provided by Vestas Wind System A/S. The rotor model of Fuga

is not compared with the other models because the chosen thrust force distribution is uniform and the total thrust force is a model input. Here, the deflections are switched off in FLEX5 and HAWC2 in order to make a fair aerodynamic comparison with EllipSys3D RANS-AD that can only model stiff blades. The near wake model of Pirrung et al. (2016, 2017) is used in HAWC2, and a uniform inflow is employed without inflow turbulence and without the presence of a wall.

The mechanical power and thrust force as function of the undisturbed wind speed are plotted in Fig. 8 for the three models;

EllipSys3D RANS-AD, FLEX5 and HAWC2. For wind speeds between 5 and 8 m/s, all three models predict a similar power and thrust coefficients that differ in order of 2%. The thrust coefficient of EllipSys3D RANS-AD and HAWC2 only differ around 1% for all wind speeds, while EllipSys3D RANS-AD overpredicts the power coefficient by about 1% below 9 m/s and 2-6% for higher wind speeds. The largest differences between FLEX5 and HAWC2 are observed around the shoulder of the power curve, which is presumably caused by differences in control strategies.

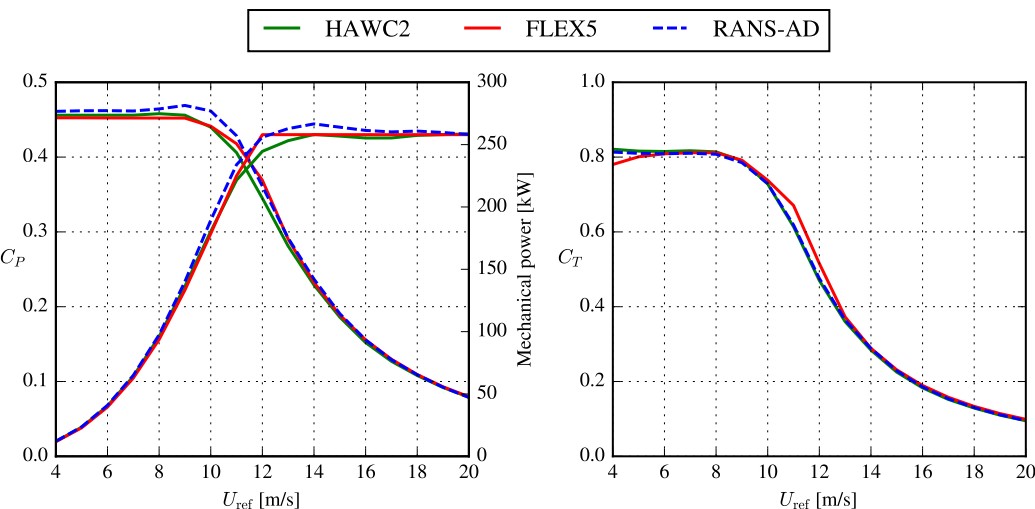

**Figure 8.** Comparison of simulated mechanical power and thrust of a single V29 rotor using HAWC2, FLEX5 and EllipSys3D RANS-AD.

The normalized tangential and thrust force distributions for three different wind speeds (7, 12 and 18 m/s) are plotted in Fig. 9 for HAWC2, FLEX5 and EllipSys3D RANS-AD. For a wind speed of 7 m/s (below rated), all three models predict similar force distributions. For the higher wind speeds (12 and 18 m/s), there are differences between the three models, mainly observed outboard and towards the blade tip, which could be related to the different tip corrections that are employed in each model.




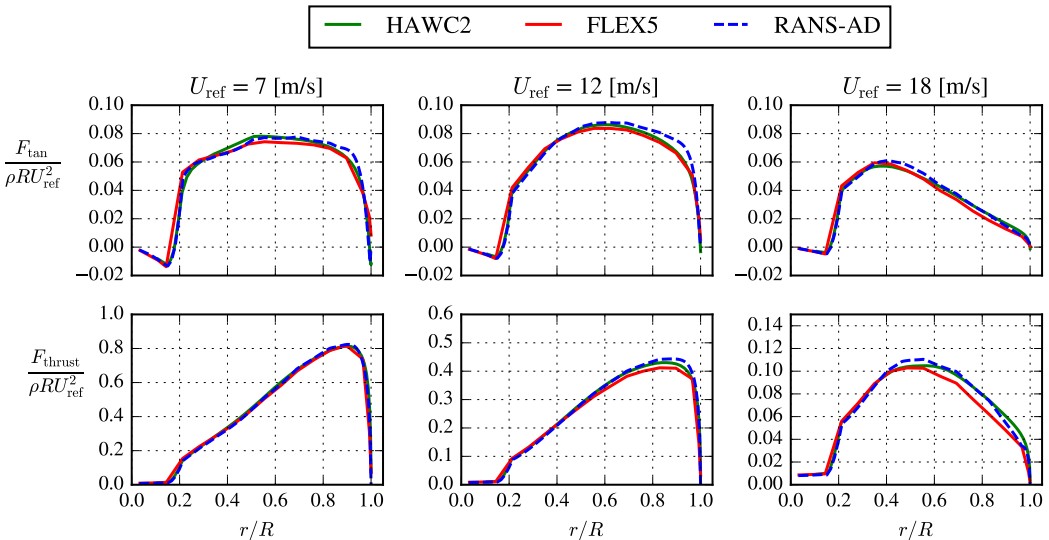

**Figure 9.** Comparison of simulated tangential (top row plots) and thrust (bottom row plots) force distribution of a single V29 rotor using
HAWC2, FLEX5 and EllipSys3D RANS-AD.

## 4.2 Performance of the 4R-V29 wind turbine

The measured and simulated relative difference in power $\Delta C_P$ and thrust force $\Delta C_T$ of the 4R-V29 wind turbine due to the
rotor interaction are depicted in Fig. 10. $\Delta C_P$ and $\Delta C_T$ are calculated as:

$$\Delta C_P = \frac{(P_{R_1}^{s_2} + P_{R_3}^{s_2}) - (P_{R_1}^{s_1} + P_{R_3}^{s_3})}{P_{R_1}^{s_1} + P_{R_3}^{s_3}} \qquad \Delta C_T = \frac{(T_{R_1}^{s_2} + T_{R_3}^{s_2}) - (T_{R_1}^{s_1} + T_{R_3}^{s_3})}{T_{R_1}^{s_1} + T_{R_3}^{s_3}} \qquad (4)$$

where $s_1$, $s_2$ and $s_3$ correspond to the three stages of the test cycle as illustrated in Fig. 2, and $P$ and $T$ are power and thrust
force for a rotor $R_i$. The measurements in the left plot of Fig. 10 show that the rotor interaction increases the power production
of the 4R-V29 wind turbine for the wind speed bins below rated between 7.5 and 11 m/s. The standard error of the mean $\Delta C_P$
is too large to make the same statement below 7.5 m/s. Above rated, the effect of the rotor interaction on the mean power is
smaller than below rated, and high uncertainties of the mean power for 11.5 and 13 m/s is observed. The weighted average of

$\Delta C_P$ (using the number of observations per bin) for a wind speed between 5 and 11 m/s is $1.8 \pm 0.2\%$, which supports the
observed bias towards a power gain below rated. The rotor interaction of the 4R-V29 wind turbine increases the annual energy
production by $1.5 \pm 0.2\%$ if we assume a Weibull distributed wind speed with shape and scale parameters of 2 and 7.5 m/s,
respectively (corresponding to a mean wind speed of about 6.7 m/s), and we assume a zero power gain below 5 m/s and above
11 m/s. The 0.2% uncertainty represents the standard error of the mean and does not represent measurement uncertainties

directly, which could be a lot higher than 0.2%. However, the analysis is focused on the relative differences between the test
cycles as illustrated from Fig. 4. In addition, we have removed uncertainties due to measurement biases as much as possible
(e.g. induction correction), as discussed in Section 2.2.





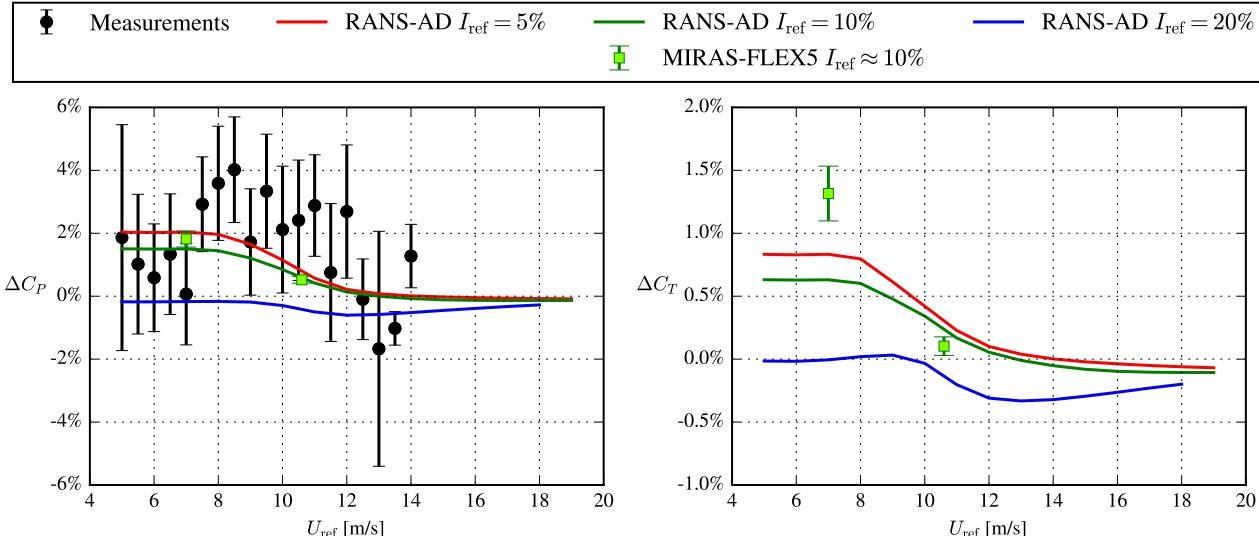

**Figure 10.** Relative difference between the 4R-V29 wind turbine with all rotors in operation and the 4R-V29 wind turbine with a single rotors in operation, in terms power and thrust as function of the freestream velocity at a height of 44.27 m.

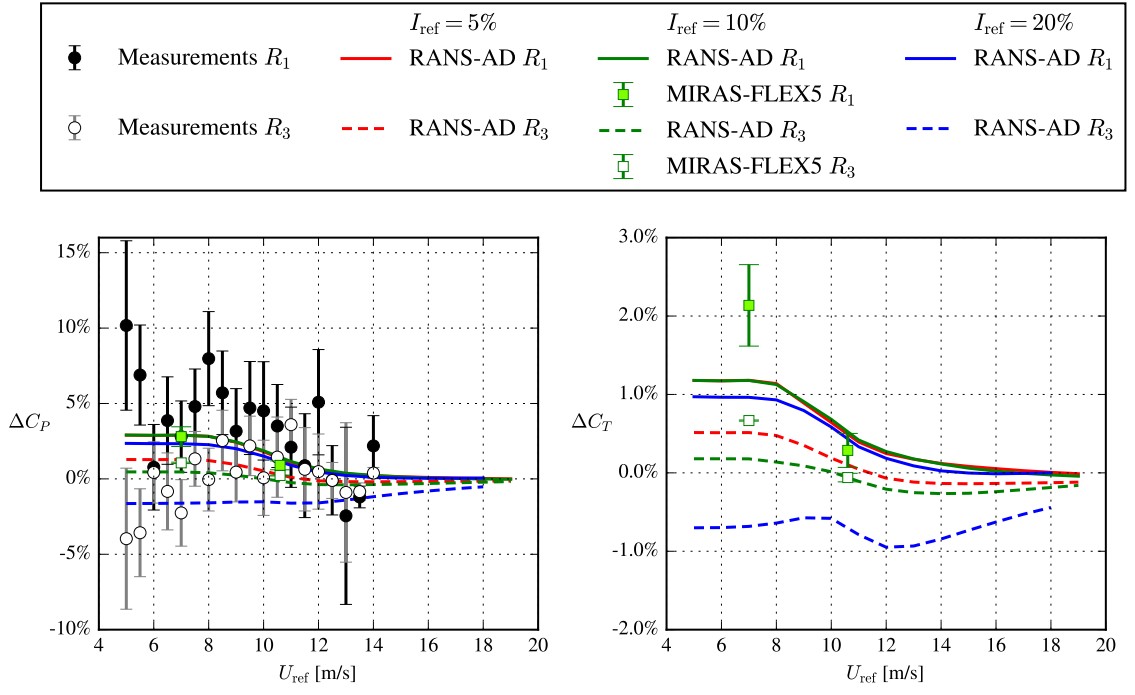

**Figure 11.** Rotor individual relative difference between the 4R-V29 wind turbine with all rotors in operation and the 4R-V29 wind turbine with a single rotors in operation, in terms of power and thrust as function of the freestream velocity at a height of 44.27 m.



The RANS-AD simulations in Fig. 10 are performed for three different turbulence intensities (5%, 10% and 20%), and a larger power and thrust force below rated is predicted when all four rotors are in operation for the two lowest turbulence intensities (5% and 10%). The largest gain in power (2%) is found for the lowest turbulence intensity, where the shear is also the lowest. For a large turbulence intensity, the effect of the rotor interaction is almost zero below rated. The loss in power

above rated power is not interesting because it possible to adapt the pitch angle such that the rated power is reached. Note that the V29 rotor starts to pitch out between 10 and 11 m/s. The right plot of Fig. 10 shows that $\Delta C_T$ from RANS-AD simulations follow the trends of the $\Delta C_P$. This indicates that the axial induction of the 4R-V29 wind turbine is increased due to the rotor interaction. The measured power gain including the standard error of the mean is in the same order as the RANS-AD simulations, except for the wind speed bins of 8.5, 11, 12 and 14 m/s, where the measured power gain is underpredicted by the

simulations. The lower measured power gain for the winds speeds below 7.5 m/s compared to wind speeds between 7.5 and 9.5 m/s can also be related to the fact that a high turbulence intensity is more common for low winds speeds, and the RANS-AD simulations show that the power gain decreases with increasing turbulence intensity.

Two results of MIRAS-FLEX5 for a wind speed of 7 and 10.6 m/s using the Mann inflow turbulence of far wake case, which has a turbulence intensity of about 10% are also depicted in Fig. 10. Each result represents the mean of two consecutive 10 min.

averages, and the error bar represents the standard error of the mean. The power gain predicted by MIRAS-FLEX5 for a wind speed of 7 and 10.6 m/s is 0.3% higher and 0.1% lower, respectively, compared to the results from RANS-AD (for a turbulence intensity of 10%); however, the trend with wind speed is the same. The gain in thrust coefficient from MIRAS-FLEX5 is 0.7% higher and 0.1% lower than RANS-AD for 7 and 10.6 m/s, respectively. The higher gains for 7 m/s from MIRAS-FLEX5 are not caused by difference in domain blockage when operating one or four rotors since effects of domain blockage are avoided,

as discussed in Section 3.3.

$\Delta C_P$ and $\Delta C_T$ for a bottom rotor ($R_1$) and a top rotor ($R_3$) calculated by the RANS-AD simulations for three different turbulence intensities are plotted in Fig. 11. The measurements in Fig. 11 also depict $\Delta C_P$ for one bottom ($R_1$) and one top rotor ($R_3$). The RANS-AD simulations indicate that the difference in $\Delta C_P$ and $\Delta C_T$ within a horizontal pair ($R_1$ compared to $R_2$ and $R_3$ compared to $R_4$) is negligible (results of $R_2$ and $R_4$ are not shown in Fig. 11 to improve readability), while

the difference between a vertical pair is clearly visible. The bottom rotors produce more $\Delta C_P$ and $\Delta C_T$ than the top rotors, and the difference between bottom and top pair increases with turbulence intensity probably due to associated increased shear. For the largest turbulence intensity (20%) and shear ($\alpha = 0.25$), only the bottom rotors produces more power, which could be caused by the difference in thrust force between the top and bottom rotors. In other words, the high thrust force of the top rotors creates a blockage effect that pushes more wind downwards into the rotor plane of the bottom rotors. Two results of

MIRAS-FLEX5, corresponding to a wind speed of 7 and 10.6 m/s and a turbulence intensity of about 10%, confirm that the bottom rotors produce more $\Delta C_P$ and $\Delta C_T$ with respect to the top rotors. In addition, the difference between MIRAS-FLEX5 and RANS-AD is the largest for the bottom rotor for 7 m/s in terms of $\Delta C_T$ (1%), where MIRAS-FLEX5 also shows the largest standard error of the mean because the lower rotor experiences a lower inflow wind speed and a higher turbulence level compared to the top rotor. The measurements also indicate that a bottom rotor is mainly responsible for the power gain,

although the standard error of the mean of the bottom and top rotor overlap for most of the wind speed bins. In addition, one





could argue that the sloping terrain, as illustrated in Fig. 5, may have influenced the difference between a top and a bottom pair, because a sloping terrain can lead to a speed up close to the ground that enhances the wind resource for the lower rotor pair. The terrain effects could be included and studied in future work.

### 4.3 Wake deficit of the 4R-V29 wind turbine

5 Results of the near wake test case are discussed in Sect. 4.3.1, while Sect. 4.3.2 presents results of the far wake test case including the near wake to far wake development.

### 4.3.1 Near wake case

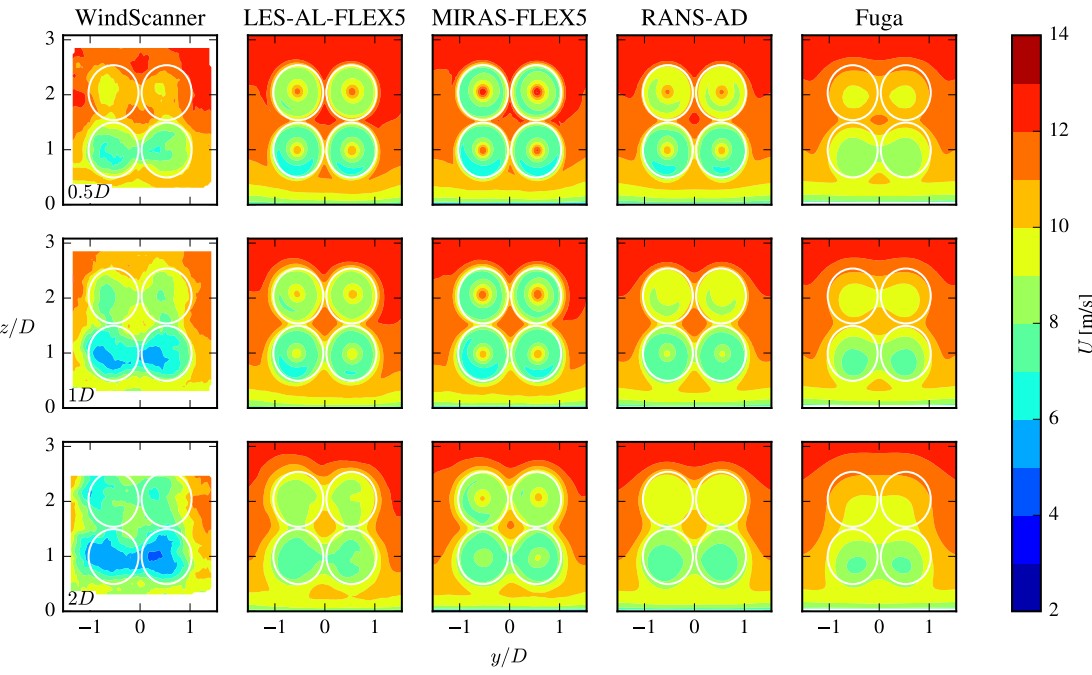

**Figure 12.** Near wake case: Contours of stream-wise velocity at three downstream distances.

Contours of the stream-wise velocity at three downstream distances, measured by the short-range WindScanner and simulated by four models (LES-AL-FLEX5, MIRAS-FLEX5, RANS-AD and Fuga) are depcited in Fig. 12. The measurements and
10 simulations show four distinct wakes, best visible at $x/D = 0.5$. At this distance, the measurements and Fuga show a stronger deficit at the bottom rotors compared to the top rotors, which also visible in the RANS-AD results with a smaller differences between the top and bottom rotors. The mixing in the surface layer is a linearly increasing with height in RANS-AD and Fuga, which increases the mixing of the top rotors compared to the bottom rotors. In the higher fidelity models: LES-AL-FLEX5 and



MIRAS-FLEX5, the inflow turbulence is modeled by Mann turbulence that has a uniform turbulent mixing in the vertical direction. This could explain why LES-AL-FLEX5 and MIRAS-FLEX5, do not show a clear difference in wake deficit between the bottom and top rotors. Note that all models include a sheared inflow, which can also cause a difference in the wake deficit between the top and bottom rotors. At $x/D = 2$, the measurements show much lower velocities compared to all four models.

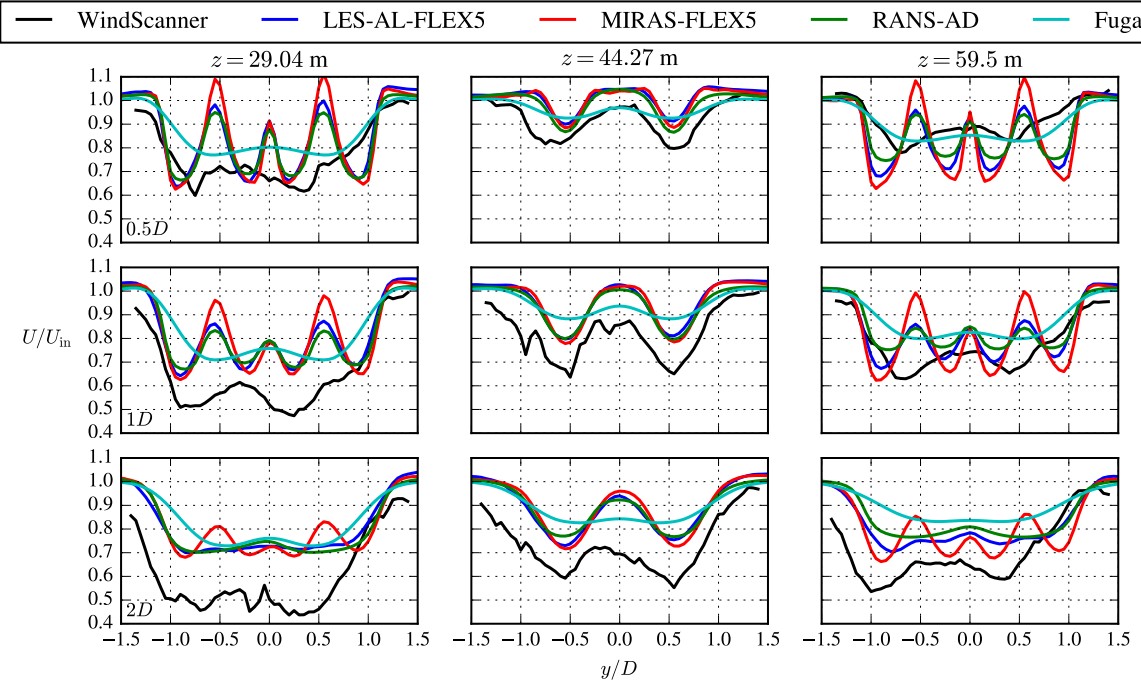

**Figure 13.** Near wake case: Profiles of stream-wise velocity at three heights and three downstream distances.

Profiles of the stream-wise velocity normalized by the inflow, at three heights, corresponding to the bottom rotor hub height (29.04 m), the center reference height (44.27 m) and the top rotor hub height (59.04 m) are plotted in Fig. 13. Results of the WindScanner and the four models are shown, taken at three downstream distances. It clear that measured wake deficit and velocity outside the wake, at the bottom rotor hub height and center height are lower than predicted by all four models. This suggest that the actual shear and reference wind speed at the 4R-V29 wind turbine could have been different from what has been measured at the reference met. mast. Unfortunately, it is not possible to determine the freestream conditions from the WindScanner data because of the limited horizontal extend of the scanned planes. In addition, the atmospheric conditions of the near wake case measured at the reference met. mast was near neutral, see Tab. 1, which could have increased measured the wake deficit.

The measurements and all models, except Fuga, show a build up of a traditional double bell shaped near wake profile at the center height when going downstream, as depicted in Fig. 13. Fuga is based on a linearized RANS approach, which means that it is designed to describe the far wake properly, while it cannot predict the nonlinear near wake accurately, especially for a high thrust coefficient, as shown by Ebenhoch et al. (2017). However, the other models yield very similar results.

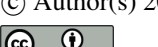



Profiles of the turbulence intensity $I$ ($I = \sqrt{2/3k}/U_{\mathrm{ref}}$) are plotted in Fig. 14 using the same definition as Fig. 13. Only results of LES-AL-FLEX5, MIRAS-FLEX5 and RANS-AD are shown because the WindScanner cannot measure $I$ and Fuga cannot model $I$ in the wake because it uses a turbulence closure that is unaffected by the wake. Fig. 14 shows that the RANS-AD has smaller peaks in $I$ with respect to LES-AL-FLEX because an AD model simulates a ring root and tip vortex while an

5  AL model resolves a (smeared) root and tip vortex per blade.

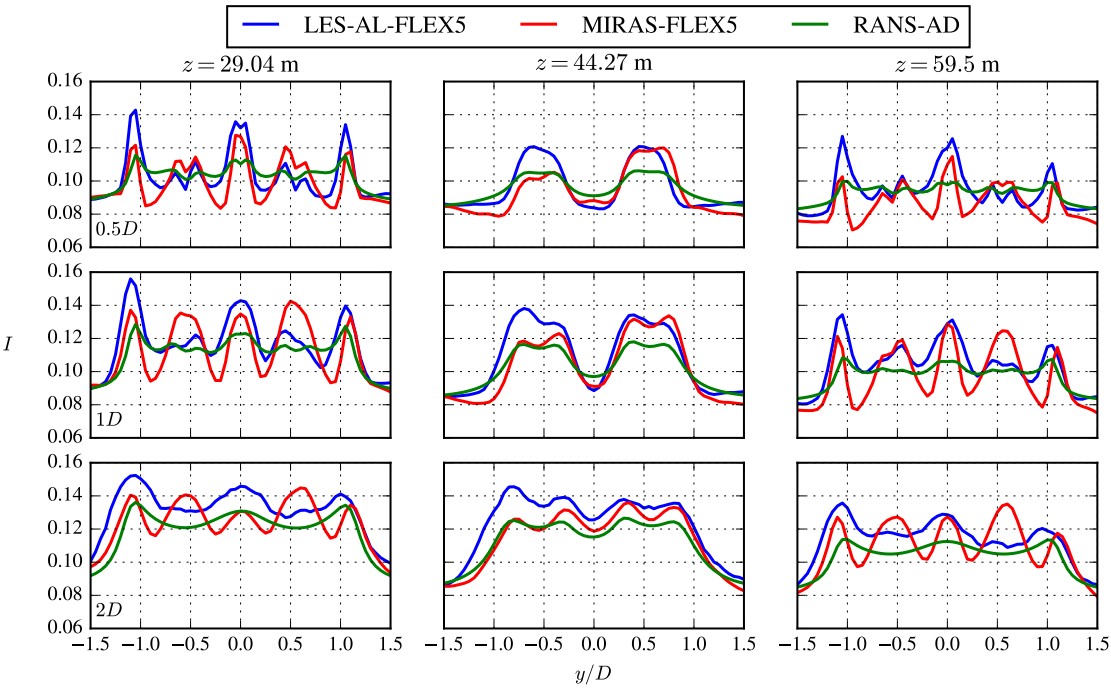

**Figure 14.** Near wake case: Profiles of turbulence intensity at three heights and three downstream distances.

### 4.3.2  Far wake case

The results of the far wakes case are plotted in Figs. 15, 16 and 17, which follow the same definition as Figs. 12, 13 and 14, respectively. In addition, six downstream distances are depicted to show the full downstream development of the 4R-V29 wind turbine wake. Only measurements of the stream-wise velocity at $x/D = 5.5$ are available. The four individual wakes merge

10  into a single structure between $x/D = 2$ and $x/D = 3$ as shown in Figs. 15 and 16. The middle column of Fig. 16 depicts how a bell shaped near wake structure forms at the center height up to and including $x/D = 3$, while the single wakes at the bottom and top hub heights cannot be distinguished from each other at this distance. Going further downstream, at $x/D = 5.5$, the fifth row of plots of Fig. 15 shows that all models capture the measured single wake structure at $x/D = 5.5$, although the wake of Fuga has moved downwards compared to the measurements and other models. The magnitude of the wake deficit at $x/D = 5.5$


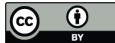

is underpredicted by all models, as seen in Fig. 16, where the measured wake at the bottom hub height is also skewed. The measured wake skewness could be a terrain effect or a results of the 8.2° yaw misalignment, as discussed in Sect. 2.3.

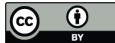

**Figure 15.** Far wake case: Contours of stream-wise velocity at three downstream distances.





**Figure 16.** Far wake case: Profiles of stream-wise velocity at three heights and six downstream distances.

The inflow Mann turbulence that is used in LES-AL-FLEX5 and MIRAS-FLEX5 results in a turbulent kinetic energy profile that has a higher value near the ground and lower value above the center height compared to the reference turbulent kinetic energy at the center height. The turbulent kinetic energy profile in the RANS-AD simulations is constant with height. Hence, the comparison of the RANS-AD simulations with LES-AL-FLEX5 and MIRAS-FLEX5 simulations in terms of turbulence intensity (Fig. 17), at $z = 29.04$ m and $z = 59.5$ m is not entirely fair. At the center height ($z = 29.04$), where the ambient turbulence intensity levels between the models are similar, the turbulence intensity in the far wake is higher in the RANS-AD simulations compared to LES-AL-FLEX5 (about 0.02 at $x/D = 12$, $y/D = 0$), which has been observed in previously



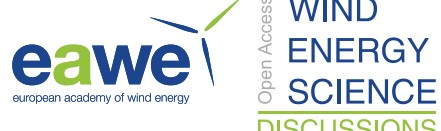

**Figure 17.** Far wake case: Profiles of turbulence intensity at three heights and six downstream distances.

by van der Laan et al. (2015) for single AD simulations. The largest difference in turbulence intensity between the LES-AL-FLEX5 and MIRAS-FLEX5 simulations are found in the near wake for the lowest rotor pair ($z = 29.04$).

The presented near and far wake cases show that the models follow the measured trends but the amount of measured data is not enough to validate the simulations. More wake measurements of the 4R-V29 wind turbine are required in order to perform

5    a model validation.





## 4.4 Wake recovery of the 4R-V29 wind turbine

The wake recovery of a multi-rotor wind turbine is very important for placing several multi-rotors together in wind farms. Therefore, the aim here is to quantify the wake recovery of a multi-rotor wind turbine operating in an atmospheric surface layer with respect to an equivalent single-rotor wind turbine that has the same rotor area, force distributions, tip speed ratio (TSR) and

total thrust force. In order to do so, a simplify the 4R-V29 wind turbine is used to provide a fair comparison with a equivalent single-rotor wind turbine can be made. The simplified 4R-V29 wind turbine has a zero toe-out angle, and the force distributions are defined by prescribed normalized blade force distributions (calculated by Réthoré et al. (2014) employing a detached eddy simulation of the NREL-5MW rotor for a wind speed of 8 m/s). The blade force distributions are scaled by the hub height velocity $U_H$, $R$, $C_T$, $C_P$ and the rotational speed (RPM) as discussed by van der Laan et al. (2015). The resulting AD force

distributions are uniform over the azimuth, and the effect of shear on the AD force distributions are neglected. The dimensions and scaling parameters of the simplified 4R-V29 wind turbine and an equivalent single-rotor wind turbine referred as V58, are summarized in Tab. 2. The inflow is an atmospheric surface layer, with $U_{\mathrm{ref}} = 7$ m/s and three different $I_{\mathrm{ref}}$: 5%, 10% and 20%, at $z_{\mathrm{ref}} = 44.27$ m. The hub height wind speed for the bottom and top rotor pairs is different for the simplified 4R-V29 wind turbine due to the shear. In other to model the same total thrust force for the V58 wind turbine, the thrust coefficient of

the V58 is adjusted. The rotational speed is set to assure a TSR of 7.6 for all rotors.

**Table 2.** Definition of the simplified 4R-V29 multi-rotor wind turbine and an equivalent V58 single-rotor wind turbine for three ambient turbulence intensities.

| $I_{\mathrm{ref}}$ | Turbine | Rotor | $D$ [m] | $z_H$ [m] | $U_H$ [m/s] | $C_T$ | $C_P$ | RPM | TSR |
|---|---|---|---|---|---|---|---|---|---|
| 5% | 4R-V29 simplified | $R_1$ and $R_2$ | 29.2 | 29.04 | 6.812 | 0.81 | 0.46 | 33.8617 | 7.6 |
| | | $R_3$ and $R_4$ | 29.2 | 59.5 | 7.132 | 0.81 | 0.46 | 35.452 | 7.6 |
| | V58 single-rotor | - | 58.4 | 44.27 | 7.0 | 0.804 | 0.46 | 17.398 | 7.6 |
| 10% | 4R-V29 simplified | $R_1$ and $R_2$ | 29.2 | 29.04 | 6.624 | 0.81 | 0.46 | 32.927 | 7.6 |
| | | $R_3$ and $R_4$ | 29.2 | 59.5 | 7.264 | 0.81 | 0.46 | 36.107 | 7.6 |
| | V58 single-rotor | - | 58.4 | 44.27 | 7.0 | 0.799 | 0.46 | 17.398 | 7.6 |
| 20% | 4R-V29 simplified | $R_1$ and $R_2$ | 29.2 | 29.04 | 6.266 | 0.81 | 0.46 | 31.148 | 7.6 |
| | | $R_3$ and $R_4$ | 29.2 | 59.5 | 7.518 | 0.81 | 0.46 | 37.373 | 7.6 |
| | V58 single-rotor | - | 58.4 | 44.27 | 7.0 | 0.792 | 0.46 | 17.398 | 7.6 |

Fig. 18 depicts the wake recovery in terms of stream-wise velocity and added turbulence intensity of the simplified 4R-V29 multi-rotor wind turbine and the equivalent V58 single-rotor wind turbine as function of stream-wise distance $x$ normalized by the single rotor diameter ($D_{\mathrm{eq}} = 58.4$ m) for three turbulence intensities (5%, 10% and 20%). The wake recovery is calculated as rotor integrated values normalized by the same integral without an AD. Note that four integrals are calculated for the multi-

rotor and summed up for each downstream distance. The left plots of Fig. 18 shows that the wake recovery distance in terms




of stream-wise velocity of a simplified 4R-V29 multi-rotor wind turbine is about $1.03\text{-}1.44D_{\text{eq}}$ shorter than the wake recovery distance of a V58 single-rotor wind turbine, which is a remarkable difference. The largest difference is found for the lowest ambient turbulence intensities (5%). This suggests that the horizontal area of a wind farm consisting of 4R-V29 wind turbines positioned in a regular rectangular layout can be reduced compared to a wind farm consisting of V58 wind turbines. The

5     area could be reduced by $1-(1-1.44/s)^2$ and $1-(1-1.03/s)^2$ (for $I_{\text{ref}} = 5\%$ and $I_{\text{ref}} = 20\%$, respectively), with $s$ as the horizontal and vertical inter turbine spacing in $D_{\text{eq}}$. For example, for $s = 8D_{\text{eq}}$ the RANS predicted reduction in wind farm area would be 24-32%, which is a significant reduction in area and hence cost or a potential increase in the power production by increasing the number of installed turbines in a given area. This result is a rough extrapolation that should be verified by wind farm simulations of multi-rotor wind turbines.

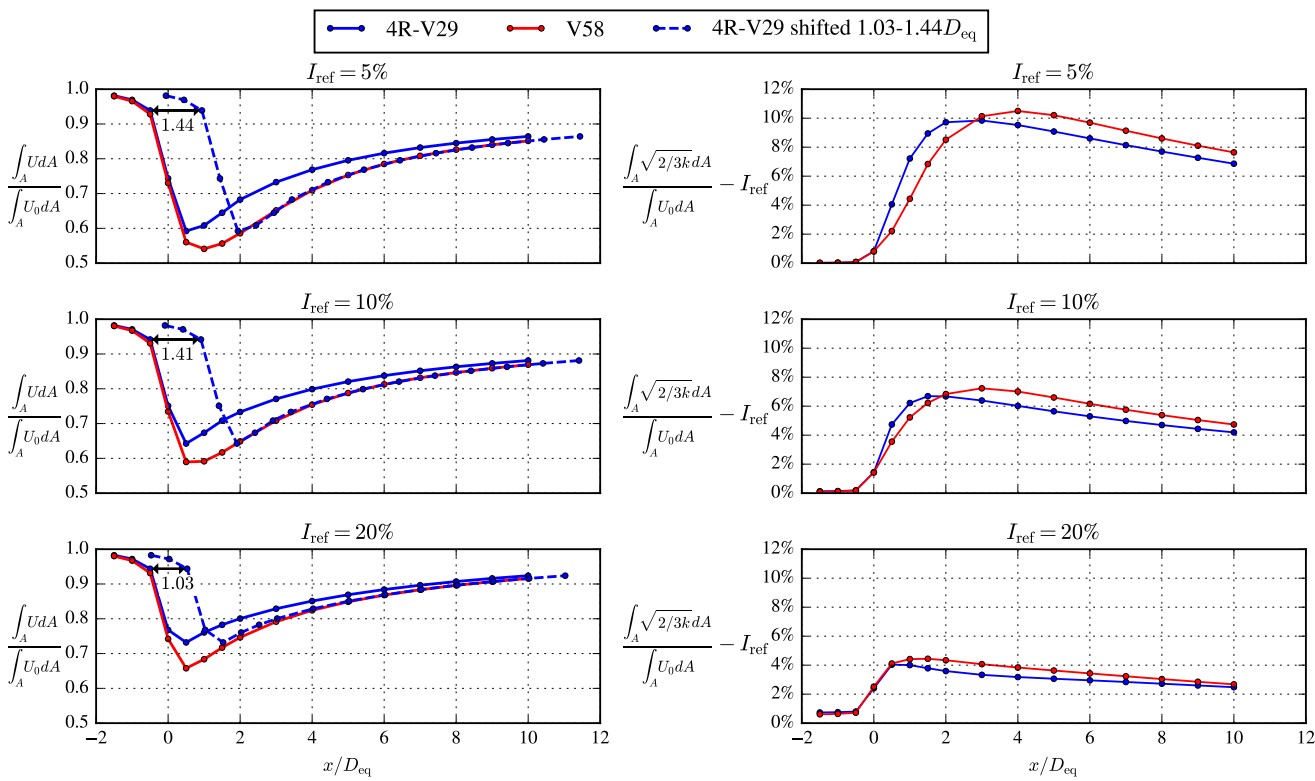

**Figure 18.** RANS predicted wake recovery of a simplified 4R-V29 multi-rotor wind turbine compared with an equivalent V58 single-rotor wind turbine for three difference turbulence intensities. Left plot: integrated stream-wise velocity, right plot: integrated added turbulence intensity. Dashed blue line is the integrated stream-wise velocity of the simplified 4R-V29 wind turbine shifted by $1.03\text{-}1.44D$.

10     The right plots of Fig. 18 show that the added wake turbulence is larger for the multi-rotor wind turbine in near wake for $I_{\text{ref}} = 5\%$ and $I_{\text{ref}} = 10\%$ for $x/D_{\text{eq}} < 3$ and $x/D_{\text{eq}} < 2$, respectively but is smaller in the far wake with respect to the added wake turbulence of single-rotor wind turbine. It is not possible to shift the added wake turbulence of the multi-rotor wind





turbine downstream to match the added wake turbulence of the single-rotor wind turbine in the same manner as the wake recovery. The lower wake turbulence in the far wake has the potential to reduce blade fatigue loads that are caused by wake turbulence.

The increased wake recovery of a multi-rotor wind turbine could be related to the fact that the total thrust force is more
distributed compared to a single-rotor wind turbine. Ghaisas et al. (2018) also obtained a faster wake recovery for a multi-rotor wind turbine and argued that it is caused by a larger entrainment because the ratio of the rotor perimeter and the rotor swept area is twice as high for the multi-rotor wind turbine with four rotors.

## 5    Conclusions

Numerical simulations and field measurements of the Vestas multi-rotor wind turbine (4R-V29) are performed. The simulations
show an increased thrust force and axial induction of the 4R-V29 wind turbine compared to a single rotor. In addition, the simulations calculate a 0-2% enhanced power performance of the 4R-V29 multi-rotor wind turbine below rated due to the interaction of the rotors. The largest gain in power is obtained for a low turbulence intensity that is associated with a low shear. The relative power gain is the largest for the bottom rotor pair. Power curve measurements of the 4R-V29 wind turbine also show that the rotor interaction increases the power performance below rated by $1.8 \pm 0.2\%$, which can result in a $1.5 \pm 0.2\%$
increase in the annual energy production, where $\pm 0.2\%$ represents the standard error of the mean.

Two flow cases based on short-range WindScanner wake measurements of the 4R-V29 wind turbine are used to compare the multi-rotor wake deficit simulated by four numerical models. In the near wake, four distinct wake deficits are visible that merge into a single structure at a downstream distance of $2-3D$. More wake measurements are required to validate the numerical models.
The wake recovery of a simplified 4R-V29 wind turbine is quantified by a comparison of the wake recovery of an equivalent single-rotor V58 wind turbine. RANS simulations show that the wake recovery distance in terms of stream-wise velocity of the simplified 4R-V29 wind turbine is 1.03-1.44$D_{eq}$ shorter than a the wake recovery distance of the equivalent single-rotor wind turbine with a rotor diameter $D_{eq}$. In addition, it is found that the added wake turbulence of the simplified 4R-V29 wind turbine is smaller than the equivalent single-rotor V58 wind turbine in the far wake. The fast wake recovery of a multi-rotor
wind turbine can potentially lead to closer spaced wind turbines in multi-rotor wind farms and needs to be further investigated.

## Appendix A:  Induction correction for the measured reference wind speed for the power curve measurements of the 4R-V29 wind turbine

The measured effect of the rotor interaction on the power production is quantified using the test cycle of Fig. 2, where the combined power curve of two single rotor operation stages (Stages 1 and 3) is compared with the power curve of a stage where
all four rotors are in operation (Stage 2) . The reference wind speed in these power curve measurements is taken at $5D$ (146 m) upstream, as discussed in Sect. 2.2. Since the induction zone of Stage 1 and 3 is smaller than Stage 2, a lower reference wind





speed is measured when all four rotors are in operation. Hence, the power curve of Stage 2 will be shifted towards the left and an artificial bias towards a power gain due to the rotor interaction would be measured. To avoid this, the reference wind speed is corrected with a factor $f_{\mathrm{cor}}$ when all four rotors are in operation (Stage 2):

$$f_{\mathrm{cor}} = \frac{\frac{1}{2}(U_{\mathrm{ref}}^{\mathrm{Stage,1}} + U_{\mathrm{ref}}^{\mathrm{Stage,3}})}{U_{\mathrm{ref}}^{\mathrm{Stage,2}}} \tag{A1}$$

for each undisturbed wind speed with an interval of 1 m/s. The induction correction factor can only be calculated if the undisturbed wind speed is known. Therefore, the RANS simulations of Sect. 3.2.2 are used to calculate $f_{\mathrm{cor}}$, and the results are shown in Fig. A1 for a reference turbulence intensity of 10%. $f_{\mathrm{cor}}$ follows the thrust coefficient curve and below rated, where the thrust coefficient is the highest, the measured reference wind speed for Stage 2 is 0.7% is lower than the reference wind speed of Stages 1 and 3.

$f_{\mathrm{cor}}$ is also calculated with a simple induction model from Troldborg and Meyer Forsting (2017), that has been developed to model the induction of a single rotor rotor in a uniform inflow. The simple induction model is only a function of the thrust coefficient, rotor radius and spatial coordinates. The thrust coefficient of the RANS simulations is used as input. The induction zone for Stage 2 is calculated by superposition of the induction of the four individual rotors. Fig. A1 shows, that the induction of the 4R-V29 wind turbine at $x = -5D$ is underestimated by the simple induction model compared to the RANS

simulations, and should not be used to correct of the reference wind speed of Stage 2. We choose to use the RANS results to correct the reference wind speed because Meyer Forsting et al. have shown that RANS-AD simulations compare well with lidar measurements of the induction zone when measurement uncertainty is included in the validation method.

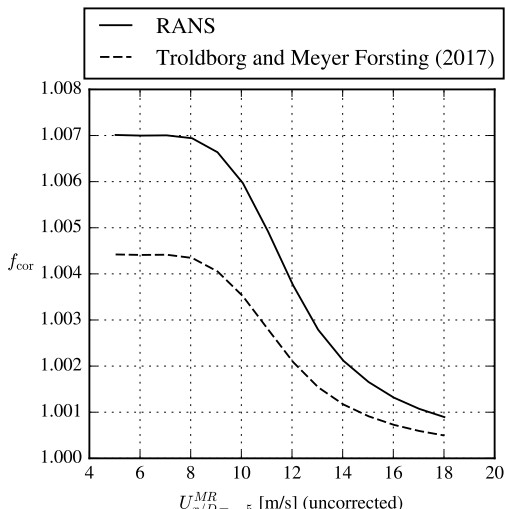

**Figure A1.** Induction correction factor for the measured reference wind speed of the 4R-V29 wind turbine.

The influence of the ambient turbulence intensity at a height of 44.27 m on $f_{\mathrm{cor}}$ in the RANS simulations is also investigated for three different turbulence intensities (5%, 10% and 20%). The results are same for a turbulence intensity of 5% and





10%, while the $f_{\mathrm{cor}}$ is slightly higher for a turbulence intensity of 20% ($f_{\mathrm{cor}} = 1.0073$ below rated). Since the power curve measurements are filtered for a wind direction from the fjord, we expect that the ambient turbulence intensity is lower than 20% and a $f_{\mathrm{cor}}$ based on a turbulence intensity of 10% is justified.

*Code and data availability.* The numerical results are generated with proprietary software, although the data presented can be made available
by contacting the corresponding author.

*Author contributions.* MPVDL has performed the EllipSys3D RANS-AD and Fuga simulations, produced all figures and drafted the article. SO, MPVDL and MK have investigated the meteorology of the wake measurements. SJA and NRG have performed the EllipSys3D LES-AD-FLEX5 and the MIRAS-FLEX5 simulations, respectively. GRP has contributed to the validation of the numerical single rotor models. NA, MS and TKM have planned, executed and post processed the WindScanner wake measurements. KHS and JXVN have executed and
10 post processed the power curve measurements. GCL has planned and managed the research related to this article. All authors jointly finalized the paper.

*Competing interests.* The authors declare that they have no conflict of interest.

*Acknowledgements.* This is work is sponsored by Vestas Wind System A/S

**Nomenclature**

$D$      Rotor diameter of each single rotor of the 4R-V29 wind turbine.
$D_{\mathrm{eq}}$      Rotor diameter of an equivalent single rotor wind turbine ($D_{\mathrm{eq}} = 2D$).




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
