# Peer review of "Power curve and wake analyses of the Vestas multi-rotor demonstrator"

_Wind Energy Science, 2018_

## Referee Comment (RC1) · Jamieson (Referee) · 30 Jan 2019

I think this paper reflects excellent experimental work.

My main concerns are with the introduction as a key aspect of the work is not recorded. The results of this paper are the first experimental corroboration in real wind operation (validation as yet would be too strong a word as the results give in effect only one data point for 4 closely spaced turbines) of power gains previously predicted by purely computational methods associated with the interaction of numbers of rotors and spacing. The main problem is that references are not up-to-date with the most substantial piece of work (Innwind task 1.33 Report of 2015) missing and papers on power enhancement from multiple turbine blockage - Nishino (2015) and others - missing.

[Figure]

Results from 2015 are;

35% power gain from an infinite array (70 actuator discs) at optimum spacing for a large array $\sim$ 1 radius (Nishino 2015); 8% from 45 rotors at minimal spacings 5% and 2.5% diameter (Innwind, Chasapogiannis, 2015); 3% from 7 rotors (cited Chasapogiannis 2014) at 5% diameter spacing.

So the 4R-V29 result of 2% from 4 rotors closely spaced fits well with the above but it is really great that is comes from real turbines operating in real turbulent wind!

The sentence starting in line 15 "it is.." etc is a misunderstanding and needs deletion or ammendment. BEM of course cannot be used in any present form for interacting rotors and all data regarding rotor interaction comes from CFD or vortex models as in Chasapogiannis. There are power gains of the Innwind 45 multi-rotor array from two separate influences - response to turbulent wind compared to a single large turbine (which does not depend on interactions - simply the faster response of small turbines modelled in BEM with structural and control system dynamics) and rotor interaction. Also the dominant effect on structure loads comparing multi and single (at least for large numbers of turbines) has nothing to do with rotor interactions but with averaging effects of many turbines.

In 5 Conclusions, p 25, line 22, a wake recovery distance of 4R-V29 of 1.03 to 1.44 Deq is "shorter" than for a single equivalent rotor. Why not approximately quantify the relevant distance for the single rotor?

Overall I think the paper should be improved by inclusion of the missing references and something like the story above regarding power gains discussed.

---

## Referee Comment (RC2) · Anonymous Referee #2 · 8 Feb 2019

The paper describes a comparison between measurements and simulations of wakes of the multi-rotor demonstrator at Risø. The paper is thorough, but unfortunately measurements and simulation results do not match up. Good to publish though, for other's to improve upon further.

Some questions:

- Does the influence of trees and highway (mentioned on page 6., line 10) perhaps explain some of the differences between measurement and simulation (far wake)?

- On page 7, line 9-11, it is stated that the atmospheric conditions were used as input to the simulations. Yet on page 18 line 11-13, it is stated that the near-neutral conditions could have reduced wake deficit of measurements relative to simulations. How can the

conditions still be "blamed" if they were taken into account in the simulations, also the more high-fidelity ones? Am I missing something?

Possible correction: - Page 18, line 7-8: Isn't wake deficit higher if wind speed is lower in the wake?

---

## Referee Comment (RC3) · Dominic von Terzi (Referee) · 21 Feb 2019

The manuscript deals with the comparison of power curve and wake data from measurements and various simulations for a multirotor-demonstrator. The authors identify a significant performance gain and the potential for a substantial denser spacing of such an architecture as compared to an equivalent single-rotor turbine. While some supporting evidence is given, I am not yet convinced of these conclusions. Nevertheless, in my view, the obtained data are of interest. Hence, I recommend to accept the manuscript for publication with minor revisions.

My two main concerns are:

1. Simulations and measurements don't seem to agree well enough and, for the far

wake, sufficient measurement data are missing (as also stated by the authors).

2. The neutral atmospheric stability conditions chosen for the simulations are not fully representative for what a turbine would see in the field and, hence, real performance gains and wake recoveries may be substantially different.

The following suggestions may help in addressing the above concerns:

- What is the estimate for measurement errors? How does this estimate change with wind speed, shear and turbulence intensity?

- Regarding figures 10 & 11, it may be useful to add lines or additional graphs for measurements filtered for similar stability conditions as in the simulations, i.e. for roughly neutral conditions or at least the same shear. If sufficient data were to remain after filtering the measurements, comparisons may be improved and the conclusions of the authors strengthened.

- I liked the concept of the equivalent single rotor used in section 4.4, but would have also liked to see a figure with power curves of this equivalent rotor compared to the (simplified) multi-rotor simulations. It may then make sense to introduce the equivalent rotor at the end of section 3 and add power curve figures after Fig. 11.

- With the concept of the equivalent single rotor, the choice of a suitable simulation tool from the authors' arsenal and systematically varying shear and turbulence intensity of the inflow, it may be possible to gain sufficient insights to explain observed power gains and accelerated wake recovery (and the differences to the measurements).

If the authors feel that the above recommendations cannot be obtained in a timely manner, they may choose to limit the manuscript to neutral stability conditions by adjusting the title and toning down the abstract and conclusions accordingly.

Additional suggestions to improve the manuscript:

- Some more proofreading may be in order, e.g. last sentence in the introduction, p. 9

line 8, captions of figures 10 & 11, p. 17 lines 11-13, p. 18 line 7, references Ghaisas et al. and Meyer et al.

- In the abstract and conclusions, providing +/-0.2% (p. 1 line4-5) may be misleading as the true measurement errors are likely substantially higher (as is stated on p. 14 lines 14-15).

- Regarding the first paragraph of the introduction, are there also some disadvantages of a multi-rotor architecture?

- Regarding figure 7, maybe plot (also) in log-log scale? Moreover, plotting spectra of longitudinal velocity fluctuations or turbulent kinetic energy of measurements and the fitted Mann model may be helpful to appreciate the quality of the inflow data.

- On p. 10 line 14 shouldn't it be C_mu=0.9 or sqrt(C_mu)=0.03?

---

## Author Comment (AC1) · 18 Mar 2019

**Reply to reviewers**

March 18, 2019

We would like to thank the three reviewers for their feedback and suggestions to improve the article. In the proceeding sections, the reviewers comments are copied and answered per comment (blue color). We have also added minor corrections and model updates ourselves:

1. We have added an earlier reference in the introduction of a multi-rotor wind mill (not a wind turbine) that was used for water management purposes in the early 1900s.

2. We have added a statement about the induction correction in Section 2.2: "It should be noted that applying the induction correction after the 10 minute averaging did not make a difference for the final power curve."

3. There was a typo in the sampling frequency of the power curve measurements, which is corrected from 20 to 1 Hz (Section 2.2).

4. There was also a typo in the legend of Figure 4: Stages 1+2 should be Stages 1+3.

5. We added a sentence in Section 3.3 to point out that both MIRAS-FLEX5 and LES-AL-FLEX5 use the same Mann turbulence scaling: "The same scaling factor is necessary in LES-AL-FLEX5 simulations, as discussed in Sect. 3.4."

An additional document is provided that highlights all modifications with respect to the initial submitted version.

**Reviewer 1 (Peter Jamieson)**

1. I think this paper reflects excellent experimental work. My main concerns are with the introduction as a key aspect of the work is not recorded. The results of this paper are the first experimental corroboration in real wind operation (validation as yet would be too strong a word as the results give in effect only one data point for 4 closely spaced turbines) of power gains previously predicted by purely computational methods associated with the interaction of numbers of rotors and spacing. The main problem is that references are not up-to-date with the most substantial piece of work (Innwind task 1.33 Report of 2015) missing and papers on power enhancement from multiple turbine blockage - Nishino (2015) and others - missing. Results from 2015 are; 35% power gain from an infinite array (70 actuator discs) at optimum spacing for a large array  1 radius (Nishino 2015); 8% from 45 rotors at minimal spacings 5% and 2.5% diameter (Innwind, Chasapogiannis, 2015); 3% from 7 rotors (cited Chasapogiannis 2014) at 5% diameter spacing. So the 4R-V29 result of 2% from 4 rotors closely spaced fits well with the above but it is really great that is comes from real turbines operating in real turbulent wind!

We have looked at the suggested additional literature:

- The final Innwind report does discuss the 45 multi-rotor wind turbine, but I could only find the 8% power increase for a tip clearance of 2.5% of the rotor diameter compared to an equivalent single-rotor wind turbine, which we have added to the introduction of the revised article.

- We have looked at the work of Nishino and Draper (2015) and we have written the following in the revised version: Nishino and Draper (2015) have employed RANS simulations of a horizontal array of ADs with a tip clearance of 50% of the rotor diameter and a constant thrust coefficient of 8/9. They find an increase in the wind farm power coefficient, based on the disc averaged velocity, up to 5%, when increasing the number of ADs from 1 to 9. Nishino and Draper (2015) have also simulated an infinite array of ADs, but the domain blockage ratio for this case was too high (2%) in order to have a valid result, as also discussed in Sect. 3.2.2 of the present article.
- The 35% power increase discussed by Nishino and Draper (2015) comes from an older article Nishino and Wilden (2012), where an array of tidal turbine in a small channel was modeled. This power increase is not relevant for the present work, because it is mainly caused by a very large blockage ratio. If one would like to model shrouded wind turbine rotors, then one could refer to the 35% power increase.
- We could not find an article of Chasapogiannis (2015).

2. The sentence starting in line 15 "it is.." etc is a misunderstanding and needs deletion or ammendment. BEM of course cannot be used in any present form for interacting rotors and all data regarding rotor interaction comes from CFD or vortex models as in Chasapogiannis.

   This has been removed.

3. There are power gains of the Innwind 45 multi-rotor array from two separate influences - response to turbulent wind compared to a single large turbine (which does not depend on interactions - simply the faster response of small turbines modelled in BEM with structural and control system dynamics) and rotor interaction. Also the dominant effect on structure loads comparing multi and single (at least for large numbers of turbines) has nothing to do with rotor interactions but with averaging effects of many turbines.

   We have added these statements to the introductions.

4. In 5 Conclusions, p 25, line 22, a wake recovery distance of 4R-V29 of 1.03 to 1.44 Deq is "shorter" than for a single equivalent rotor. Why not approximately quantify the relevant distance for the single rotor?

   The wake recovery of the equivalent single-rotor wind turbine is compared with a simplified multi-rotor wind turbine. We could define a wake recovery distance at which the wake is 85% recovered (for example) for both wind turbines. We then find similar numbers (approximately) as presented in the article because we can shift the velocity deficit of the multi-rotor wind turbine such that it matches the velocity deficit of the single-rotor wind turbine.

5. Overall I think the paper should be improved by inclusion of the missing references and something like the story above regarding power gains discussed.

   See first answer.

**Reviewer 2 (Anonymous)**

The paper describes a comparison between measurements and simulations of wakes of the multi-rotor demonstrator at Risø. The paper is thorough, but unfortunately measurements and simulation results do not match up. Good to publish though, for other's to improve upon further.

1. Does the influence of trees and highway (mentioned on page 6., line 10) perhaps explain some of the differences between measurement and simulation (far wake)?

   Yes, this could have influenced the far wake results. We have now added a small statement in the Results and Discussion section (Section 4.3.2) and we have added that it is challenging to compare the models with a single 8 minute averaged result of the WindScanner.

2. On page 7, line 9-11, it is stated that the atmospheric conditions were used as input to the simulations. Yet on page 18 line 11-13, it is stated that the near-neutral conditions could have reduced wake deficit of measurements relative to simulations. How can the conditions still be "blamed" if they were taken into account in the simulations, also the more high-fidelity ones? Am I missing something?

   This has been clarified in Section 2.3. We employ neutral atmospheric conditions for all models.

3. Possible correction: - Page 18, line 7-8: Isn't wake deficit higher if wind speed is lower in the wake?

   We have corrected this.

**Reviewer 3 (Dominic von Terzi)**

The manuscript deals with the comparison of power curve and wake data from measurements and various simulations for a multirotor-demonstrator. The authors identify a significant performance gain and the potential for a substantial denser spacing of such an architecture as compared to an equivalent single-rotor turbine. While some supporting evidence is given, I am not yet convinced of these conclusions. Nevertheless, in my view, the obtained data are of interest. Hence, I recommend to accept the manuscript for publication with minor revisions. My two main concerns are:

1. Simulations and measurements don't seem to agree well enough and, for the far wake, sufficient measurement data are missing (as also stated by the authors).

   We agree. This is why an additional measurement campaign has been carried at the end of 2018, which focuses on the far wake velocity deficit of the multi-rotor wind turbine using a long-range lidar system. We plan to present these additional measurements in a follow up article.

2. The neutral atmospheric stability conditions chosen for the simulations are not fully representative for what a turbine would see in the field and, hence, real performance gains and wake recoveries may be substantially different.

   You are right that the effect of atmospheric stability is important. Some of the employed simulations tools can include these effects, although they still need a more thorough validation, which could be done in another article dedicated to the effect atmospheric stability. In the present work, we have chosen to restrict ourselves to (near) neutral conditions. However, the employed RANS-AD model has been used with different ambient turbulence intensities (5%, 10% and 20%) corresponding to wide range of wind shears (shear parameters of 0.06, 0.13 and 0.25, respectively). The influence of the ambient turbulence intensity on the power gain between is significant and shows that the effect of different atmospheric conditions is important.

3. The following suggestions may help in addressing the above concerns: What is the estimate for measurement errors? How does this estimate change with wind speed, shear and turbulence intensity?

   This is good and relevant question and we have had related discussion among the authors. We are aware that the uncertainty or bias of the individual absolute power measurement may be higher than the measured power gain. However, we have tried to minimize the effect of this by using consecutive power measurements to quantify the rotor interaction: Stage 1 (1 bottom rotor $R_1$ in operation for 15 min.), Stage 2 (All rotors in operation for 15 min. and only use power measurements of $R_1$ and $R_3$), Stage 3 (1 top rotor $R_3$ in operation for 15 min), as discussed in Sections 2.2 and 4.2 In addition, we have corrected for the difference in the induction, as discussed in Appendix A.

4. Regarding figures 10 11, it may be useful to add lines or additional graphs for measurements filtered for similar stability conditions as in the simulations, i.e. for roughly neutral conditions or at least the same shear. If sufficient data were to remain after filtering the measurements, comparisons may be improved and the conclusions of the authors strengthened.

   We agree. However, the amount of data is not large enough to allow additional filtering. We have added to Section 2.1: "In addition, the power curve measurements are not filtered for turbulence intensity and atmospheric stability, because the amount of data after filtering would be too small."

5. I liked the concept of the equivalent single rotor used in section 4.4, but would have also liked to see a figure with power curves of this equivalent rotor compared to the (simplified) multi-rotor simulations. It may then make sense to introduce the equivalent rotor at the end of section 3 and add power curve figures after Fig. 11.

   I think there is a misunderstanding of the RANS-AD setup used to simulate a simplified multi-rotor and an equivalent single-rotor wind turbine. In order to make a fair comparison of the wake deficit, we have chosen to define thrust and tangential force distributions and scale it with the relevant parameters (as mentioned in the article), such that both the total thrust and tip speed ratio is the same for the multi-rotor and equivalent single-rotor wind turbines. This mean that AD model of the simplified setup cannot be used to calculate the wind turbine power, because it is an input rather than a result. Note that input $C_P$ curve of the both wind turbines is the same. This would mean that the power curve of the equivalent single rotor is the same as the power curve of the multi-rotor wind turbine.

6. With the concept of the equivalent single rotor, the choice of a suitable simulation tool from the authors' arsenal and systematically varying shear and turbulence intensity of the inflow, it may be possible to gain sufficient insights to explain observed power gains and accelerated wake recovery (and the differences to the measurements).

   We have continued the concept of the equivalent single rotor in a recently submitted wake conference article using both RANS and LES. This work is focused on multi-rotor wind farms.

7. If the authors feel that the above recommendations cannot be obtained in a timely manner, they may choose to limit the manuscript to neutral stability conditions by adjusting the title and toning down the abstract and conclusions accordingly.

   This is a bit difficult, because the power curve measurements do include effects of stability, while the wake measurements are near neutral and the simulations are neutral. Hence we cannot add to the title that we only consider neutral or near neutral conditions. However, the text clearly explains the stability conditions per measurement campaign and simulation setup.

8. Additional suggestions to improve the manuscript: Some more proofreading may be in order, e.g. last sentence in the introduction, p. 9 line 8, captions of figures 10 11, p. 17 lines 11-13, p. 18 line 7, references Ghaisaset al. and Meyer et al.

   Corrected. We could not find a typo in the reference to Ghaisas et al.

9. In the abstract and conclusions, providing +/-0.2% (p. 1 line4-5) may be misleading as the true measurement errors are likely substantially higher (as is stated on p. 14 lines 14-15).

   We understand this concern and we have had similar discussions among the authors. We have now decided to remove the reported standard error of the mean in the conclusion and abstract, while we keep them in the Results and Discussion section where the representation of the $\pm 0.2\%$ is explained.

10. Regarding the first paragraph of the introduction, are there also some disadvantages of a multi-rotor architecture?

    Yes, one could argue that more components are needed and a more complex tower is required. We have added some discussion to the introduction.

11. Regarding figure 7, maybe plot (also) in log-log scale? Moreover, plotting spectra of longitudinal velocity fluctuations or turbulent kinetic energy of measurements and the fitted Mann model may be helpful to appreciate the quality of the inflow data.

    We have considered to add plots of the fitted Mann spectra, but we decided to leave them out of the article to limit the number of pages. Regarding Figure 7, we prefer to use a linear scale for the height because the turbulence intensity is not linear in a log scale.

12. On p. 10 line 14 shouldn't it be $C_\mu = 0.9$ or $\sqrt{(C_\mu)} = 0.03$?

    The standard value of $C_\mu$ is the $k$-$\varepsilon$ model is 0.09. However, for atmospheric flows, it is common to use $C_\mu = 0.03$, which is based on surface layer measurements taken at several places, see for example Page 28 of the PhD thesis of the main author van der Laan (2014):

The value of $C_\mu$ is based on the log law solution for $k$:

$$C_\mu = \frac{u_*^4}{k^2} = \frac{u_*^4}{\frac{1}{4}\left(\sigma_u^2 + \sigma_v^2 + \sigma_w^2\right)^2} = 0.03, \tag{1}$$

where $\sigma_u$, $\sigma_v$ and $\sigma_w$ are measured standard deviations of the velocity components in a neutral atmospheric surface layer, that have been summarized by Panofsky and Dutton (1984):

$$\frac{\sigma_u}{u_*} = 2.39 \pm 0.03, \qquad \frac{\sigma_v}{u_*} = 1.92 \pm 0.05, \qquad \frac{\sigma_w}{u_*} = 1.25 \pm 0.03. \tag{2}$$

[revised manuscript text omitted]